# INTERACTION FIELD MATCHING: OVERCOMING LIMITATIONS OF ELECTROSTATIC MODELS

**Stepan Manukhov**[*]
MSU[†] Faculty of Physics, Moscow, Russia
Applied AI Institute, Moscow, Russia
manukhov2000akk@gmail.com

**Alexander Kolesov***
Applied AI Institute, Moscow, Russia
AXXX, Moscow, Russia
alexander.kolesov@gmail.com

**Vladimir V. Palyulin**
Applied AI Institute, Moscow, Russia
v.palyulin@gmail.com

**Alexander Korotin**
Applied AI Institute, Moscow, Russia
AXXX, Moscow, Russia
iamalexkorotin@gmail.com

## ABSTRACT

Electrostatic field matching (EFM) has recently appeared as a novel physics-inspired paradigm for data generation and transfer using the idea of an electric capacitor. However, it requires modeling electrostatic fields using neural networks, which is non-trivial because of the necessity to take into account the complex field outside the capacitor plates. In this paper, we propose Interaction Field Matching (IFM), a generalization of EFM which allows using general interaction fields beyond the electrostatic one. Furthermore, inspired by strong interactions between quarks and antiquarks in physics, we design a particular interaction field realization which solves the problems which arise when modeling electrostatic fields in EFM. We show the performance on a series of toy and image data transfer problems. Our code is available at https://github.com/justkolesov/InteractionFieldMatching.

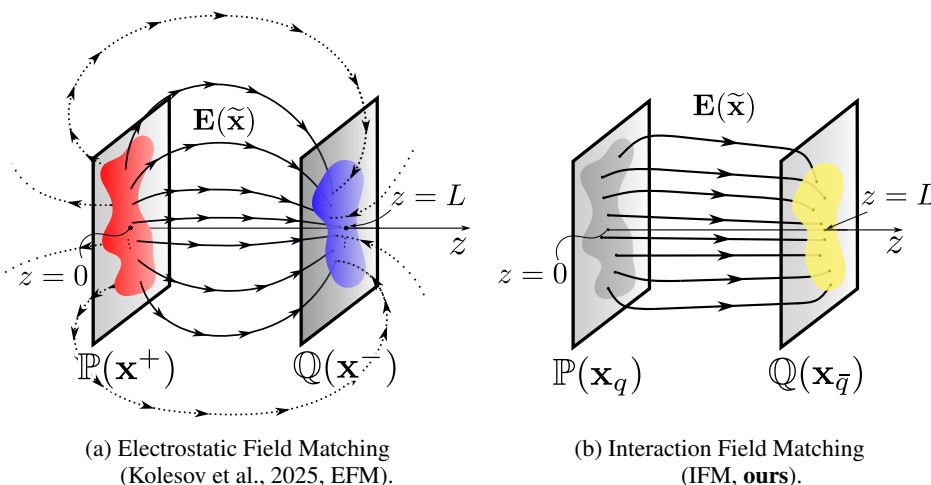

(a) Electrostatic Field Matching
(Kolesov et al., 2025, EFM).

(b) Interaction Field Matching
(IFM, **ours**).

Figure 1: Electrostatic Field Matching (Kolesov et al., 2025, EFM) and our Interaction Field Matching (IFM) concepts. Two $D$-dimensional distributions $\mathbb{P}(\cdot)$, $\mathbb{Q}(\cdot)$ are placed in $\mathbb{R}^{D+1}$ at $z = 0$ and $z = L$ **(a)** In EFM, the distributions are interpreted as charges creating a capacitor-like electric field. Movement along these field lines transfers the distributions, but requires consideration of all directions of the field lines. **(b)** Our IFM is a generalization of the EFM to arbitrary interactions between charges. One possible realization of IFM is motivated by the strong interaction between quarks. This realization does not have backward-oriented lines and has a smaller curvature of lines.

---

[*]Equal contribution.
[†]Moscow State University

# 1 INTRODUCTION

While diffusion (Sohl-Dickstein et al., 2015; Ho et al., 2020) and flow matching (Liu et al., 2023; Lipman et al., 2023; Albergo & Vanden-Eijnden, 2023) models dominate current research in deep generative modeling, a new paradigm grounded in Coulomb electrostatics has emerged (Xu et al., 2022; Kolesov et al., 2025; Cao & Zhao, 2024; Cao et al., 2024; Xu et al., 2023). Early work in this direction introduced Poisson Flow Generative Models (Xu et al., 2022; 2023, PFGM), focusing on noise-to-data generation. More recently, Electrostatic Field Matching (Kolesov et al., 2025, EFM) generalized this framework, enabling electrostatic models to solve data-to-data transfer problems.

Electrostatic Field Matching (EFM) draws inspiration from electric capacitors, modeling input and target distributions as positive and negative electrostatic charges, respectively. The method performs distribution transfer by following electrostatic field lines (Fig. 1a). While conceptually simple, EFM faces significant practical challenges: it requires accounting for all field lines—including *backward-oriented* ones (dotted lines in Fig. 1a)—which exhibit high curvature and span the entire space. This makes them difficult to model, as the necessary training volume becomes unbounded.

In this paper, we tackle the limitations of EFM (§2.3) and deliver the following **main contributions**:

1. **Theory.** We propose Interaction Field Matching (IFM), a generic paradigm for distribution transfer rooted in pairwise interactions between particles from input and target distributions (§3.3, 3.5). Compared to EFM that relies on the electrostatic field, our approach allows us to leverage *general interaction fields* (beyond the Coloumb electrostatics) that satisfy certain physics-inspired properties such as the flux conservation and the *generalized* superposition principle (§3.2).

2. **Methodology & practice.** Inspired by the *strong interaction* of quarks and antiquarks in physics (§3.1), we design a particular realization of the interaction field (§3.4) which has several preferable properties compared to the electrostatic field: **(a)** the field lines have almost straight segments, **(b)** the field vanishes outside the area between particles and **(c)** it allows using the Minibatch Optimal Transport Pooladian et al. (2023) to enforce certain properties on the transfer map.

We showcase the performance of IFM on a series of toy and image data transfer problems (§4).

# 2 BACKGROUND AND RELATED WORKS

In this section, we first recall the concepts of the basic high-dimensional electrostatic (§2.1). Then we discuss its application to generative modeling and data transfer problems using the example of EFM (§2.2). Finally, in §2.3, we discuss the limitations of EFM which motivated our study.

## 2.1 ELECTROSTATICS

We recall the fundamental principles of electrostatics necessary for understanding electrostatic-based generative models. A detailed treatment of three-dimensional electrostatics can be found in any standard electricity and magnetism textbook, e.g., (Landau & Lifshitz, 1971, Chapter 5). The generalization of electrostatics to high-dimensional spaces is discussed in (Caruso et al., 2023).

**The electrostatic field.** Let $q : \mathbb{R}^D \to \mathbb{R}$ be the density of a charge distribution on $\mathbb{R}^D$. The distribution may contain both positive and negative charges and is assumed to have finite total charge ($\int |q(\mathbf{x})| d\mathbf{x} < \infty$). At a point $\mathbf{x} \in \mathbb{R}^D$ it produces the electrostatic field $\mathbf{E} : \mathbb{R}^D \to \mathbb{R}^D$:

$$\mathbf{E}(\mathbf{x}) = \int \frac{1}{S_{D-1}} \frac{(\mathbf{x} - \mathbf{x}')}{||\mathbf{x} - \mathbf{x}'||_2^D} q(\mathbf{x}') d\mathbf{x}', \tag{1}$$

where $S_{D-1}$ is the surface area of an $(D-1)$-dimensional sphere with unit radius. That is, the field at $\mathbf{x}$ is a weighted sum of fields from all charges $\mathbf{x}'$, where closer charges yield stronger field.

**Electric field strength lines.** An electric field strength line is a curve $\mathbf{x}(\tau) \in \mathbb{R}^D$, $\tau \in [a, b] \subset \mathbb{R}$ whose tangent to each point is parallel to the electric field at that point. In other words:

$$\frac{d\mathbf{x}(\tau)}{d\tau} = \mathbf{E}(\mathbf{x}). \tag{2}$$

Electric field lines are a key concept for electrostatic generative models such as PFGM and EFM.

## 2.2 ELECTROSTATIC FIELD MATCHING (EFM)

The first application of electrostatics to generative modeling problems was carried out in the works of (Xu et al., 2022; 2023, PFGM), where the authors proposed a model applicable to noise-to-data generative problems. Electrostatic Field Matching (EFM) extends the application of electrostatics to the case of data-to-data transfer, and uses previously unconsidered properties of electric field lines. We describe here EFM since it is more general than PFGM, and our work is built upon it.

EFM works with two data distributions $\mathbb{P}(\mathbf{x}^+)$ and $\mathbb{Q}(\mathbf{x}^-), \mathbf{x}^{\pm} \in \mathbb{R}^D$. The first distribution is assigned a positive charge, while the second distribution is assigned a negative charge. The distributions are placed in the *extended space* $\mathbb{R}^{D+1}$ on the planes $z = 0$ and $z = L$, respectively (see Fig. 1a). One can think of it as a $(D + 1)$-dimensional **capacitor**. A point in this space has the form $(x_1, x_2, ..., x_D, z) = (\mathbf{x}, z) = \widetilde{\mathbf{x}} \in \mathbb{R}^{D+1}$. The field is found from the superposition principle:

$$\mathbf{E}(\widetilde{\mathbf{x}}) = \mathbf{E}_+(\widetilde{\mathbf{x}}) + \mathbf{E}_-(\widetilde{\mathbf{x}}), \tag{3}$$

where $\mathbf{E}_+(\mathbf{x})$ and $\mathbf{E}_-(\widetilde{\mathbf{x}})$ are the fields created by $\mathbb{P}(\widetilde{\mathbf{x}}^+)$ and $\mathbb{Q}(\widetilde{\mathbf{x}}^-)$, respectively.

Then, as proved in the original paper, the **movement along the field lines** $d\widetilde{\mathbf{x}} = \mathbf{E}(\widetilde{\mathbf{x}})d\tau$ **performs the transfer** between the distributions $\mathbb{P}(\widetilde{\mathbf{x}}^+)$ and $\mathbb{Q}(\widetilde{\mathbf{x}}^-)$. This fact has opened the possibility to use electrostatics both in data generation and transfer problems. Indeed, to move between data distributions, it is sufficient to follow the electric field lines.

To obtain a distribution transfer model, one trains a neural network $f_\theta(\cdot) : \mathbb{R}^{D+1} \to \mathbb{R}^{D+1}$ to recover the normalized electric field $\frac{\mathbf{E}(\widetilde{\mathbf{x}})}{||\mathbf{E}(\widetilde{\mathbf{x}})||_2}$, e.g., by using a loss function

$$\mathbb{E}_{\widetilde{\mathbf{x}}}||f_\theta(\widetilde{\mathbf{x}}) - \frac{\mathbf{E}(\widetilde{\mathbf{x}})}{||\mathbf{E}(\widetilde{\mathbf{x}})||_2}||_2 \to \min_\theta. \tag{4}$$

Here, $\mathbf{E}(\widetilde{\mathbf{x}})$ is calculated with (3), where $\mathbf{E}^{\pm}(\widetilde{\mathbf{x}})$ is approximated by empirical samples of $\mathbb{P}(\widetilde{\mathbf{x}}^+)$ and $\mathbb{Q}(\widetilde{\mathbf{x}}^-)$, i.e., data. Monte Carlo averaging $\mathbb{E}_{\widetilde{\mathbf{x}}}$ is done on the points $\widetilde{\mathbf{x}}$ around the plates. This set of points if called the **training volume**; its selection is crucial but highly non-trivial (Xu et al., 2022).

## 2.3 LIMITATIONS OF EFM

Despite its performance, EFM has a few weak spots coming from the properties of electrostatic fields:

**1. Backward-oriented field lines.** Each plate produces two sets of electric field lines (Fig. 1a). The first set (*forward-oriented* lines) is directed toward the second plate. The second set (*backward-oriented* lines) starts from the first plate in the opposite direction. In practice, the forward-oriented

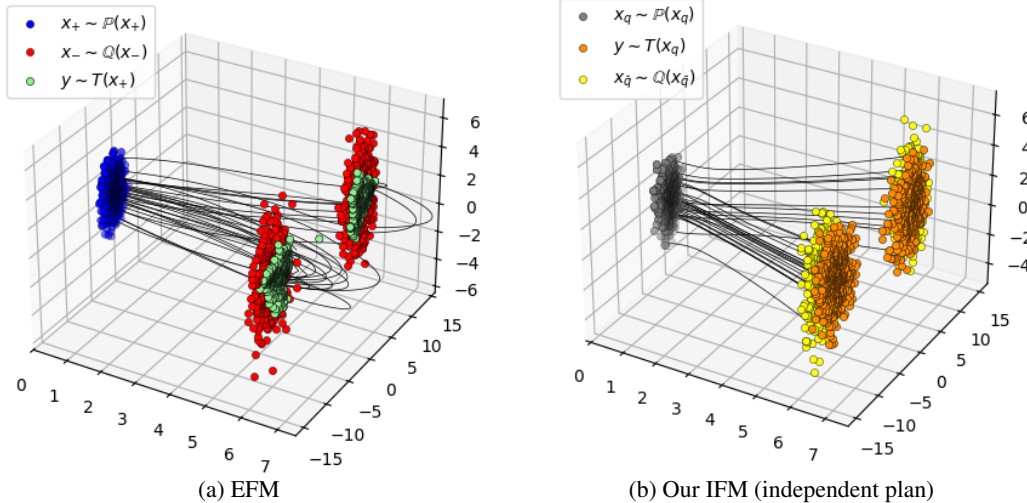

(a) EFM           (b) Our IFM (independent plan)

Figure 2: Limitations of the EFM & comparison with IFM. **(a)** The toy experiment ($1 \to 2$ Gaussians) shows that even some *forward-oriented* field lines can leave $z > L$. These trajectories have increased length and curvature. Moreover, the transfer along only the forward-oriented lines does not cover the target distribution (green point cloud does not coincide with the red one). **(b)** Our realization of IFM (§3.4) does not have the above mentioned problems: the field lines between the planes are almost straight, they do not extend beyond $z > L$ and are enough to cover the entire target distribution.

set of lines is chosen because it requires less training volume and because these lines are less curved than the lines of the backward-oriented series. However, backward-oriented lines play a critical role for the full coverage of the target distribution. The use of only forward-oriented lines is not sufficient to fully cover the distribution of $\mathbb{Q}(\cdot)$, see the illustration in Fig.2a.

**2. Line termination problem.** Even some forward-oriented field lines can pass the boundary $z = L$ before reaching the second distribution. In such a case, the field line enters the region $z > L$ (see Fig. 2a) and requires further integration to come back to target distribution at $z = L$. This problem complicates the data transfer procedure. Indeed, one has to design some criterion to decide whether the line terminates at $z = L$ or should be integrated further.

**3. Training volume selection.** From the first two problems follows the challenge of choosing the training volume, i.e., points $\widetilde{\mathbf{x}}$ in equation 4 for learning the field. For the correct transport between $\mathbb{P}(\cdot)$ and $\mathbb{Q}(\cdot)$, it is necessary to know not only the field between the plates ($0 < z < L$), but also beyond the plates ($z > L$ for lines leaving the boundary and $z < 0$ for backward-oriented lines). Therefore, it is necessary to choose a large training volume for learning of the neural network. Moreover, the size of the required volume is initially unknown.

Below we propose a generalization of EFM which aims to ease the above-mentioned problems.

# 3 INTERACTION FIELD MATCHING (IFM)

This section describes our proposed Interaction Field Mathcing (IFM), the generalization of the electrostatic paradigm in generative models. In §3.1, we start by motivating the IFM with the strong interaction between subnuclear particles (quarks) in physics. In §3.2, we present the necessary requirements for an interaction field required for our ideas to work. In §3.3, we formulate the main theorem devoted to transfer of distributions into each other by means of a proper interaction field. The §3.4 describes a particular realization of the field inspired by strong interactions. In §3.5, we report the learning and inference algorithms. The proofs are in Appendix A.1.

## 3.1 MOTIVATION: STRONG INTERACTION IN PHYSICS

To address EFM challenges, we propose utilizing the **strong interaction** (Quevedo & Schachner, 2024, §7.4)— a fundamental force binding subnuclear particles. The smallest particles involved in this interaction are called **quarks**.

A typical configuration of the strong field is shown in Fig. 3, highlighting key contrasts with electromagnetic fields. At small distances, quark $q$ and antiquark $\overline{q}$ interact similarly to charged particles $q_{\pm}$. However, as the separation increases, the strong field lines become considerably straighter.

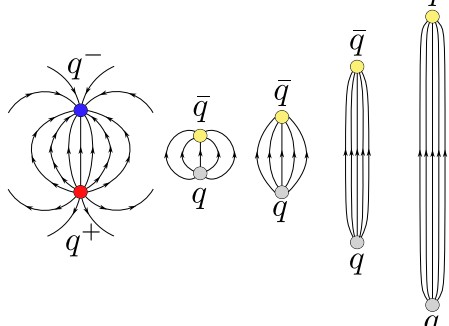

Unfortunately, strong field strength calculation requires complex quantum-mechanical computations. Although our work is motivated by quark interactions, unlike EFM and PFGM, *our setting uses modified physical interactions*.

Figure 3: Comparison of electrostatic interaction between charges $q^{\pm}$ (left) and strong interaction between quarks $q, \overline{q}$ (right). At small distances, the strong interaction resembles the electromagnetic interaction, but as quarks separate, the field lines straighten into a string.

## 3.2 PROPERTIES OF PROPER INTERACTION FIELDS

Here we list the most general requirements for the interaction field $\mathbf{E}(\widetilde{\mathbf{x}})$ which are sufficient to perform data transfer. These requirements allow for broad flexibility in the field design. In particular, it could be an electrostatic field (see Example 3.2 below). Nevertheless, to preserve the concept of strong interaction, we will still refer to particles as quarks and antiquarks.

Suppose that a quark $q$ is located at the point $\widetilde{\mathbf{x}}_q \in \mathbb{R}^{D+1}$ and an antiquark $\overline{q}$ at the point $\widetilde{\mathbf{x}}_{\overline{q}} \in \mathbb{R}^{D+1}$ and produce interation field $\mathbf{E}(\widetilde{\mathbf{x}}) = E(\widetilde{\mathbf{x}}) \cdot \mathbf{n}(\widetilde{\mathbf{x}})$, where $\mathbf{n}(\widetilde{\mathbf{x}})$ is the unit vector tangent to the field line and $E(\widetilde{\mathbf{x}})$ is the magnitude. We require the following properties of the interaction field $\mathbf{E}(\widetilde{\mathbf{x}})$:

**1. The start and the termination of lines at (anti)quarks.** For $q\bar{q}$-pair with equal charges, the interaction field line must start at the quark and end at the antiquark:

$$\begin{cases} \frac{d\widetilde{\mathbf{x}}(\tau)}{d\tau} = \mathbf{n}\big(\widetilde{\mathbf{x}}(\tau)\big), \\ \widetilde{\mathbf{x}}(\tau_s) = \widetilde{\mathbf{x}}_q, \quad \widetilde{\mathbf{x}}(\tau_f) = \widetilde{\mathbf{x}}_{\bar{q}}, \end{cases} \quad (5)$$

where $\tau_s, \tau_f$ correspond to the initial and final points of the field line.

**2. Flux conservation.** For a $q\bar{q}$-pair with equal charges, an interaction field must maintain the following property along the stream tube

$$\mathbf{E}(\widetilde{\mathbf{x}}) \cdot \mathbf{dS} = \text{const}, \quad (6)$$

where $\mathbf{dS}$ is a *vector* with a length equal to the area $dS$ of the surface, where the considered stream tube rests. The direction of the vector is orthogonal to the surface. In turn, $\mathbf{E} \cdot \mathbf{dS} = E dS \cos\alpha = E_1 dS_1 +$

Figure 4: An illustration of the flux conservation.

$... + E_D dS_D$ denotes the inner product between the vectors $\mathbf{E}$ and $\mathbf{dS}$. Informally, this property means that the number of field lines along the stream surface is constant, see Fig. 4. We additionally assume that the *total* flux between quark-antiquark pair is proportional to the charge of the quark $q$ that creates this field, and does not depend on the relative position of the quark-antiquark pair.

**3. Generalized superposition principle w.r.t. a transport plan**. Consider two continuous distributions $q(\cdot), \bar{q}(\cdot)$ of quarks and antiquarks, respectively, and assume they have the same total charge. Let $\pi(\cdot, \cdot)$ be a transport plan between these distributions, i.e., it satisfies the non-negativity property $\pi(\mathbf{x}_q, \mathbf{x}_{\bar{q}}) \geq 0$ and the marginal constraints $\int \pi(\mathbf{x}_q, \mathbf{x}_{\bar{q}}) d\mathbf{x}_{\bar{q}} = q(\mathbf{x}_q)$, $\int \pi(\mathbf{x}_q, \mathbf{x}_{\bar{q}}) d\mathbf{x}_q = \bar{q}(\mathbf{x}_{\bar{q}})$. Let $\mathbf{E}_{\mathbf{x}_q, \mathbf{x}_{\bar{q}}}(\widetilde{\mathbf{x}})$ denote the field produced by a pair of a unit quark and a unit antiquark located at $\mathbf{x}_q, \mathbf{x}_{\bar{q}}$. Then the field of the system of quarks $q$ and antiquarks $\bar{q}$ w.r.t. $\pi$ is given by:

$$\mathbf{E}_\pi(\widetilde{\mathbf{x}}) = \iint \pi(\mathbf{x}_q, \mathbf{x}_{\bar{q}}) \mathbf{E}_{\mathbf{x}_q, \mathbf{x}_{\bar{q}}}(\widetilde{\mathbf{x}}) d\mathbf{x}_q d\mathbf{x}_{\bar{q}}. \quad (7)$$

Physically, it means that $\mathbf{E}$ is the average of fields of interacting pairs $(\mathbf{x}_q, \mathbf{x}_{\bar{q}})$, and $\pi(\mathbf{x}_q, \mathbf{x}_{\bar{q}})$ characterizes the strength of pairing between quarks $\mathbf{x}_q$ and $\mathbf{x}_{\bar{q}}$. For completeness, we note that the superposition principle can be analogously stated for the discrete systems of quarks.

Properties 1-2 are formulated for a quark-antiquark pair. However, if these properties are true for the $q\bar{q}$-pair, they remain valid for more complex discrete and continuous systems.

**Lemma 3.1** (On the field lines)**.** *Let $q(\cdot)$ and $\bar{q}(\cdot)$ be two compactly supported (discrete or continuous) distributions of quarks and antiquarks. Let them satisfy $\int q(\mathbf{x}) d\mathbf{x}_q = \int \bar{q}(\mathbf{x}_{\bar{q}}) d\mathbf{x}_{\bar{q}}$. Let the field of the quark-antiquark pair $\mathbf{E}_{\mathbf{x}_q, \mathbf{x}_{\bar{q}}}(\widetilde{\mathbf{x}})$ start at $\mathbf{x}_q$ and terminate at $\mathbf{x}_{\bar{q}}$ (Property 1), and conserve flux along the current tube (Property 2). Then the total field (7) from all quarks and antiquarks*

*(a) Start at supp($\mathbb{P}$) and end at supp($\mathbb{Q}$), except perhaps for the number of lines of zero flux.*

*(b) Conserve flux along the current tubes.*

**Example 3.2** (The electrostatic field)**.** The electrostatic field (3) is a special case of the interaction field satisfying properties 1-3 for an **arbitrary** transport plan. Indeed, property 1 corresponds to the dipole field, property 2 follows from Gauss's theorem (Kolesov et al., 2025, §2.1), and property 3 is easy to check:

$$\mathbf{E}_\pi(\widetilde{\mathbf{x}}) = \int \big(\mathbf{E}_{q^+}(\widetilde{\mathbf{x}}, \widetilde{\mathbf{x}}^+) + \mathbf{E}_{q^-}(\widetilde{\mathbf{x}}, \widetilde{\mathbf{x}}^-)\big) \pi(\widetilde{\mathbf{x}}^+, \widetilde{\mathbf{x}}^-) d\widetilde{\mathbf{x}}^+ d\widetilde{\mathbf{x}}^- =$$

$$= \int \mathbf{E}_{q^+}(\widetilde{\mathbf{x}}, \widetilde{\mathbf{x}}^+) q^+(\widetilde{\mathbf{x}}^+) d\widetilde{\mathbf{x}}^+ + \int \mathbf{E}_{q^-}(\widetilde{\mathbf{x}}, \widetilde{\mathbf{x}}^-) |q^-|(\widetilde{\mathbf{x}}^-) d\widetilde{\mathbf{x}}^- = \mathbf{E}^+(\widetilde{\mathbf{x}}) + \mathbf{E}^-(\widetilde{\mathbf{x}}) = (3), \quad (8)$$

where $\mathbf{E}_q(\widetilde{\mathbf{x}}, \widetilde{\mathbf{x}}^\pm)$ is the electric field at a point $\widetilde{\mathbf{x}}$ produced by a point charge $q = \pm 1$ located at a point $\widetilde{\mathbf{x}}^\pm$. *Electrostatic fields are independent* of the transport plan $\pi$, i.e., changing the plan does not alter the field as the total field of two charges itself is a sum of two separate fields by these charges.

### 3.3 MAIN THEOREM

Let $\mathbb{P}(\mathbf{x}_q)$ and $\mathbb{Q}(\mathbf{x}_{\bar{q}})$ be two $D$-dimensional data distributions. Similarly to EFM, we put these distributions in the *extended space* $\mathbb{R}^{D+1}$ on the planes $z = 0$ and $z = L$, respectively, see Fig. 1b. Now assume that $\mathbb{P}$ and $\mathbb{Q}$ are the distributions of quarks $q$ and antiquarks $\bar{q}$, respectively. Fix a transport plan $\pi$ between them, e.g., set the independent one $\pi = q \times \bar{q} = \mathbb{P} \times \mathbb{Q}$. Let $\mathbf{E}_\pi(\widetilde{\mathbf{x}})$ be a proper interaction field, i.e., satisfying properties 1-3 of §3.2.

For transport between the distributions, we define a map $T$ : $\text{supp}(\mathbb{P}) \to \text{supp}(\mathbb{Q})$ that moves along the field lines by integrating $d\widetilde{\mathbf{x}} = \mathbf{E}(\widetilde{\mathbf{x}})d\tau$, where $\mathbf{E}(\widetilde{\mathbf{x}})$ is defined by (7). Field lines starting on the distribution $\mathbb{P}(\mathbf{x}_q)$ can emanate in two directions: forward-oriented, directed toward $\mathbb{Q}(\mathbf{x}_{\bar{q}})$, and backward-oriented, pointed initially in the opposite direction (see Fig. 5). The choice between these directions of motion must be made stochastically. A definition of the stochastic map $T$ is provided in Appendix A.2.

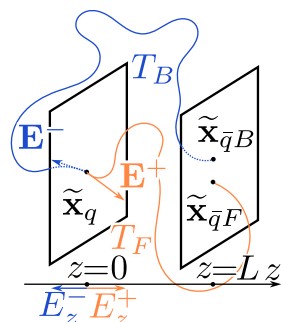

Then, for this map $T(\mathbf{x}_q)$, we prove the following key theorem:

**Theorem 3.3** (Interaction Field Matching). *Let $\mathbb{P}(\boldsymbol{x}_q)$ and $\mathbb{Q}(\boldsymbol{x}_{\bar{q}})$ be two continuous data distributions that have compact support. Let $\boldsymbol{x}_q$ be a random variable distributed as $\mathbb{P}(\boldsymbol{x}_q)$. Then $\boldsymbol{x}_{\bar{q}} = T(\boldsymbol{x}_q)$ is a random variable distributed as $\mathbb{Q}(\boldsymbol{x}_{\bar{q}})$:*

Figure 5: Illustration of forward, backward lines.

$$\boldsymbol{x}_q \sim \mathbb{P}(\boldsymbol{x}_q) \Rightarrow T(\boldsymbol{x}_q) = \boldsymbol{x}_{\bar{q}} \sim \mathbb{Q}(\boldsymbol{x}_{\bar{q}}). \tag{9}$$

In other words, the movement along interaction field lines provably transfers $\mathbb{P}(\mathbf{x}_q)$ to $\mathbb{Q}(\mathbf{x}_{\bar{q}})$. The proof of the theorem is given in Appendix A.3.

### 3.4 REALIZATION OF THE INTERACTION FIELD

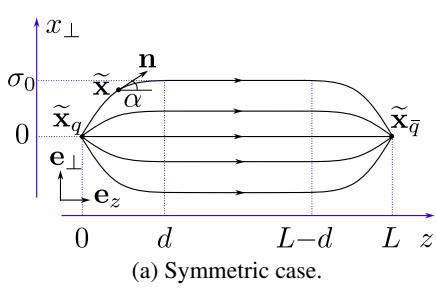
(a) Symmetric case.

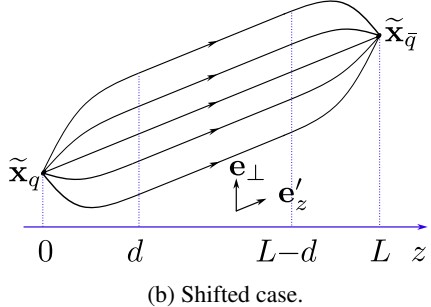
(b) Shifted case.

Figure 6: Realization of the field between two quarks. In the range $z \in [0, d]$ and $z \in [L - d, L]$ the field lines curve toward the quarks, in the middle range $z \in [d, L - d]$ the field lines are straight. The string has an effective width $\sigma_0$, beyond which the field value exponentially decreases. **(a)** Symmetric case. **(b)** The shifted case is obtained by a proportional shift along the plane $z = L$

We present an interaction field realization that meets Properties 1-3 (§3.2), motivated by quark interactions (§3.1). This design eliminates backward-oriented lines and prevents field lines from going to $z > L$ (Fig. 2b). A schematic is shown in Fig. 6, while the detailed algorithm for calculating the field is formulated in Appendix A.4.

**Theorem 3.4** (Properties of our interaction field). *Our realization of the interaction field $\mathbf{E}(\widetilde{\mathbf{x}})$ satisfies the fundamental Properties 1-3 in §3.2, with additional characteristics:*

- *Field lines never extend beyond $z > L$.*

- *No backward-oriented lines exist.*

The combination of these properties saves a given interaction field from EFM problems (§2.3). We give proof and provide additional discussions in App. A.5.

### 3.5 LEARNING AND INFERENCE ALGORITHM

To move between data distributions, it is sufficient to follow the interaction field lines. The lines can be found from the trained neural net approximating the interaction field $\mathbf{E}(\widetilde{\mathbf{x}})$.

**Training.** To recover the interaction field $\mathbf{E}(\cdot)$ in $(D + 1)$-dimensional points between the hyperplanes, similarly to EFM, we approximate it with a neural network $f_\theta(\cdot) : \mathbb{R}^{D+1} \to \mathbb{R}^{D+1}$. To begin

with, we need to define the training volume, i.e., the procedure to sample $\widetilde{\mathbf{x}}$ for training. We use the following sampling scheme:

$$\widetilde{\mathbf{x}} = (1 - \frac{z}{L})\widetilde{\mathbf{x}}_q + \frac{z}{L}\widetilde{\mathbf{x}}_{\bar{q}} + \widetilde{\epsilon}(z), \tag{10}$$

where $(\widetilde{\mathbf{x}}_q, \widetilde{\mathbf{x}}_{\bar{q}}) \sim \pi$ is sampled from the plan, $z \sim r(z)$ is the schedule distribution on $[0, L]$ and $\widetilde{\epsilon}(z)$ is the amount of noise injected into the linear interpolation of $\widetilde{\mathbf{x}}_q$ and $\widetilde{\mathbf{x}}_{\bar{q}}$ at level $z$. This scheme is inspired by related works Kolesov et al. (2025); Xu et al. (2022). In generation experiments (§4.2), we use a uniform distribution for $r(z)$ and set $\widetilde{\epsilon}(z)$ equal to zero, because the linear interpolation of a data point $\widetilde{\mathbf{x}}_q$ and a noise sample $\widetilde{\mathbf{x}}_{\bar{q}}$ already includes noise. In translation experiments (§4.3), we use also uniform $r(z)$ and $\mathcal{N}(0, \sigma^2(z))$ for $\widetilde{\epsilon}(z)$, where the variance $\sigma^2(z)$ is $= \frac{L}{2} - |\frac{L}{2} - z|$. This formulation ensures that $\widetilde{\epsilon}(z) = 0$ at $z = 0$ and $z = L$, with the maximum noise occurring at the middle $z = \frac{L}{2}$. Also, it is worth noticing that (10) is just one of many possible ways to define intermediate points between the plates, and our method does not have direct connection to the train-

---

**Algorithm 1** IFM Training

**Input:** Distributions accessible by samples:
  $\mathbb{P}(\mathbf{x}_q)\delta(z)$ and $\mathbb{Q}(\mathbf{x}_{\bar{q}})\delta(z - L)$;
  Transport plan $\pi(\mathbf{x}_q, \mathbf{x}_{\bar{q}}) : \mathbb{R}^D \times \mathbb{R}^D \to \mathbb{R}$;
  NN approximator $f_\theta(\cdot) : \mathbb{R}^{D+1} \to \mathbb{R}^{D+1}$;
**Output:** The learned interaction field $f_\theta(\cdot)$
**Repeat until converged:**
  Sample $|B|$ batch $(\widetilde{\mathbf{X}}_q, \widetilde{\mathbf{X}}_{\bar{q}}) \sim \pi(\mathbf{x}_q, \mathbf{x}_{\bar{q}})$
  Sample $|B|$ coordinates $z \sim r(z)$ ;
  Compute noise $\widetilde{\epsilon}(z)$ as $\epsilon\sigma(z)$ (see App.A.4)
  Calculate batch $\widetilde{\mathbf{x}} = (1 - \frac{z}{L})\widetilde{\mathbf{x}}_q + \frac{z}{L}\widetilde{\mathbf{x}}_{\bar{q}} + \widetilde{\epsilon}(z)$;
  Calculate $\mathbf{E}_{q\bar{q}}(\widetilde{\mathbf{x}})$ for all pairs following §3.4;
  Calculate $\mathbf{E}(\widetilde{\mathbf{x}})$ with (7)
  Compute $\mathcal{L} = \mathbb{E}_{\widetilde{\mathbf{x}}}||f_\theta(\widetilde{\mathbf{x}}) - \frac{\mathbf{E}(\widetilde{\mathbf{x}})}{||\mathbf{E}(\widetilde{\mathbf{x}})||_2}||_2^2$;
  Optimize $\min_\theta \mathcal{L}$;
  Update $\theta$ by using $\frac{\partial \mathcal{L}}{\partial \theta}$;

---

ing procedures of popular Flow Lipman et al. (2023); Liu et al. (2023); Albergo & Vanden-Eijnden (2023) or Bridge Matching Shi et al. (2023).

The ground-truth $\mathbf{E}(\widetilde{\mathbf{x}})$ is estimated with Eq. (7). Specifically, we approximate the field by Monte Carlo samples from the given transport plan $\pi$. The field between paired quarks in a batch $\mathbf{E}_{\mathbf{x}_q, \mathbf{x}_{\bar{q}}}$ is determined according to the recipe described in §3.4. Analogously to EFM and PFGM, We learn $f_\theta(\cdot)$ by minimizing the squared error difference between the normalized ground truth $\mathbf{E}(\widetilde{\mathbf{x}})$ and the predictions $f_\theta(\widetilde{\mathbf{x}})$ over the parameters of the neural network with SGD, i.e., the learning objective is

$$\mathbb{E}_{\widetilde{\mathbf{x}}}||f_\theta(\widetilde{\mathbf{x}}) - \frac{\mathbf{E}(\widetilde{\mathbf{x}})}{||\mathbf{E}(\widetilde{\mathbf{x}})||_2}||_2^2 \to \min_\theta . \tag{11}$$

**Inference.** After learning the normalized vector field $\frac{\mathbf{E}(\cdot)}{||\mathbf{E}(\cdot)||}$ with a neural network $f_\theta(\cdot)$, we simulate the movement between hyperplanes to transfer data from $\mathbb{P}(\mathbf{x}_q)$ to $\mathbb{Q}(\mathbf{x}_{\bar{q}})$. A straightforward approach for this is to run an ODE solver for equation 2. However, one needs a right stopping time for the ODE solver since the arrival time may differ for different field lines. In order to find it, we follow the idea of (Xu et al., 2022; Kolesov et al., 2025) and use an equivalent ODE solver with $\widetilde{\mathbf{x}}$ evolving with the extended variable z:

$$d\widetilde{\mathbf{x}} = (\frac{d\mathbf{x}}{dt}\frac{dt}{dz}, 1)dz = (\mathbf{E}_x(\widetilde{\mathbf{x}})\mathbf{E}_z^{-1}(\widetilde{\mathbf{x}}), 1)dz =$$
$$= (\frac{\mathbf{E}_x(\widetilde{\mathbf{x}})}{||\mathbf{E}(\widetilde{\mathbf{x}})||}\frac{||\mathbf{E}(\widetilde{\mathbf{x}})||}{\mathbf{E}_z(\widetilde{\mathbf{x}})}, 1)dz \approx (\frac{f_\theta(\widetilde{\mathbf{x}})_x}{f_\theta(\widetilde{\mathbf{x}})_z}, 1)dz, \tag{12}$$

where we denote $f_\theta(\widetilde{\mathbf{x}})$ as $(f_\theta(\widetilde{\mathbf{x}})_x, f_\theta(\widetilde{\mathbf{x}})_z)$ and $\mathbf{E}(\widetilde{\mathbf{x}})$ equals $(\mathbf{E}_x(\widetilde{\mathbf{x}}), \mathbf{E}_z(\widetilde{\mathbf{x}}))$. In the new ODE (12), we replace the time variable $t$ with the physically meaningful variable $z$ setting the explicit start $(z = 0)$ and the end $(z = L)$ conditions. We start with samples from $\mathbb{P}(\mathbf{x}_q)$, i.e., when $z = 0$. Then, we arrive at the data distribution $\mathbb{Q}(\mathbf{x}_{\bar{q}})$ when $z$ reaches $L$ during the ODE simulation.

---

**Algorithm 2** IFM Sampling

**Input:** samples $\tilde{\mathbf{x}}_q$ from $\mathbb{P}(\mathbf{x}_q)\delta(z)$; step size $\Delta\tau > 0$;
  The learned field $f_\theta^*(\cdot) : \mathbb{R}^{D+1} \to \mathbb{R}^{D+1}$;
**Output:** samples $\tilde{\mathbf{x}}_{\bar{q}}$ from $\mathbb{Q}(\mathbf{x}_{\bar{q}})\delta(z - L)$
Set $\mathbf{x}_0 = \tilde{\mathbf{x}}_q$
**for** $\tau \in \{0, \Delta\tau, 2\Delta\tau, \dots, L - \Delta\tau\}$ **do**
  Calculate $f_\theta^*(\mathbf{x}_\tau) = (f_\theta^*(\mathbf{x}_\tau)_x, f_\theta^*(\mathbf{x}_\tau)_z)$
  $\widetilde{\mathbf{x}}_{\tau+\Delta\tau} \leftarrow [(\widetilde{\mathbf{x}}_\tau)_x + f_\theta^*(\widetilde{\mathbf{x}}_\tau)_z^{-1}f_\theta^*(\widetilde{\mathbf{x}}_\tau)_x\Delta\tau; \tau + \Delta\tau]$
$\widetilde{\mathbf{x}}^{\bar{q}} \leftarrow \widetilde{\mathbf{x}}_L$

---

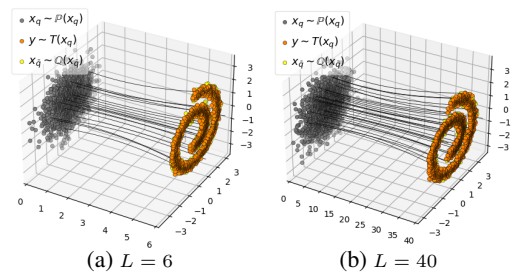

(a) $L = 6$       (b) $L = 40$

Figure 7: Interaction field line structure for the Gaussian→Swiss Roll experiment with $L = 6, 40$. Our IFM with minibatch OT plan.

We emphasize, that *in EFM such a movement does not always realize transport between distributions properly* due to the backward-oriented lines and line termination problem (§2.3). In contrast, in our implementation of IFM (§3.4), such integration theoretically provably translates $\mathbb{P}$ to $\mathbb{Q}$. All the ingredients for training and inference in our method are described in Algorithms 1 and 2, where we summarize the learning and the inference procedures.

## 4 EXPERIMENTAL ILLUSTRATIONS

Here we show the proof-of-concept experiments with our IFM method. We show a 2-dimensional illustrative experiment (§4.1), image generation (§4.3) and image-to-image translation experiments (§4.2). We give technical details of the implementation of the experiments in Appendix B. We provide a study of the sensitivity of our model to the choice of hyperparameters in Appendix C.

### 4.1 GAUSSIAN TO SWISS ROLL

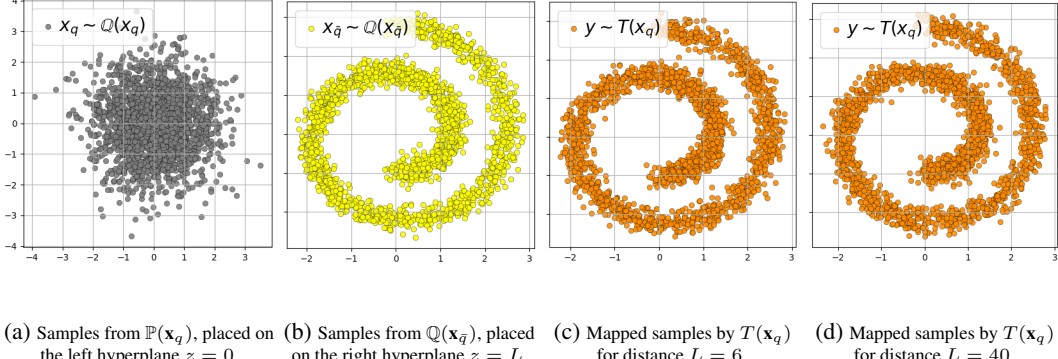

(a) Samples from $\mathbb{P}(\mathbf{x}_q)$, placed on the left hyperplane $z = 0$.

(b) Samples from $\mathbb{Q}(\mathbf{x}_{\bar{q}})$, placed on the right hyperplane $z = L$.

(c) Mapped samples by $T(\mathbf{x}_q)$ for distance $L = 6$.

(d) Mapped samples by $T(\mathbf{x}_q)$ for distance $L = 40$.

Figure 8: *Illustrative 2D Gaussian→Swiss Roll*: Input and target distributions $\mathbb{P}(\mathbf{x}_q)$ and $\mathbb{Q}(\mathbf{x}_{\bar{q}})$ with the transfer results learned by our IFM (with minibatch OT plan) for distances $L = 6$ and $L = 40$.

An intuitive initial test to validate the method involves transferring between distributions with visually comparable densities. We use a 2D zero-mean Gaussian distribution with identity covariance matrix as $\mathbb{P}(\mathbf{x}_q)$ and a Swiss Roll distribution as $\mathbb{Q}(\mathbf{x}_{\bar{q}})$. Their respective visualizations appear in Figs. 8a and 8b. In Figs. 8c and 8d show the points $T(\mathbf{x}_q)$ obtained by moving along the lines of our interaction field realization with minibatch OT plan (Tong et al., 2023). Fig. 8c corresponds to the plate distance $L = 6$, and Fig. 8d corresponds to $L = 40$. It can be seen that there are no significant differences due to the choice of the hyperparameter $L$, while in EFM the choice of large $L$ lead to failure on the same experiment, namely, it failed to accurately map $\mathbb{P}$ to $\mathbb{Q}$ see (Kolesov et al., 2025, §4.2). Fig. 7 shows the $3D$ structure of the field lines for $L = 6$ and $L = 40$. It can be seen that the lines are almost straight over the entire range of values and depend weakly on $L$. Note that in EFM the field lines were significantly curved at large values of $L$ (Kolesov et al., 2025, §5), see Fig. 7.

### 4.2 IMAGE GENERATION

We consider the generative task on the multimodal $32 \times 32$ CIFAR-10 dataset and the high-dimensional $64 \times 64$ CelebA faces dataset. In this experiment, we place noise images from a $D$-dimensional multivariate normal distribution $\mathcal{N}(0, I_{D \times D})$ on the left hyperplane $z = 0$, where $D = 3 \times 32 \times 32$ for CIFAR-10 and $D = 3 \times 64 \times 64$ for CelebA. Images from the CIFAR-10 and CelebA datasets are placed on the right hyperplane $z = 20$. In accordance with our Algorithm 1, we learn the normalized interaction field between the hyperplanes using independent transport plans.

For completeness, we compare our approach not only with previous electrostatic-based approaches such as EFM (Kolesov et al., 2025), PFGM(Xu et al., 2022) and PFGM++ (Xu et al., 2023), but also with modern flow-based FM (Lipman et al., 2023), diffusion-based DDPM (Ho et al., 2020), and adversarial approaches such as StyleGAN (Karras et al., 2020). Our method, IFM, performs competitively with well-known state-of-the-art approaches in terms of qualitative results (see Figs. 9a and 9b), while EFM fails to generate samples for the $64 \times 64$ CelebA dataset (see Fig. 9b). We quantitatively evaluate our method's performance by reporting FID in Table 1.

| Dataset / Method | IFM (our) | EFM | PFGM++ | PFGM | FM | DDPM | StyleGAN |
|---|---|---|---|---|---|---|---|
| CIFAR-10 (32x32) | 2.28 | 2.62 | 2.15 | 2.76 | 2.99 | 3.12 | 2.48 |
| Celeba (64x64) | 3.07 | >100 | 2.89 | 3.95 | 14.45 | 12.26 | 3.68 |

Table 1: *Image Generation:* FID↓ score on 32×32 CIFAR-10 and 64×64 Celeba faces datasets for our **IFM**, **EFM**, **PFGM** & **PFGM++**, flow matching (**FM**), diffusion (**DDPM**) and **StyleGAN**.

Additional qualitative results for other image generation tasks (128×128 CelebA dataset and conditional image generation on CIFAR-10) are provided in Appendices D and E

**Computational efficiency.** Training of our IFM takes less than 10 hours on a single NVIDIA A100 GPU (30 GB VRAM) for the 32×32 and 64×64 resolution datasets, and less than 30 hours for the 128×128 resolution dataset. Our IFM shares the same architecture as the closest competitors: EFM, PFGM/PFGM++, DDPM, and FM. We also use the same Euler-based ODE solver and 100 evaluation steps for each method to ensure a fair comparison. Therefore, the inference speed and memory usage is identical for all these methods. For completeness, we report the inference speed to generate different batch sizes of images, including a single image (i.e., batch size=1), see Table 2. In particular, the peak GPU memory usage for all methods is approximately 8, 10 and 16 GB during inference with a batch size of 128 for 32x32, 64x64 and 128x128 datasets, respectively.

| Dataset / Batch Size | 256 | 128 | 64 | 16 | 1 |
|---|---|---|---|---|---|
| CIFAR-10 (32x32) | 10.93 | 5.74 | 1.63 | 0.82 | 0.7 |
| Celeba (64x64) | 36.81 | 18.45 | 8.5 | 2.93 | 0.97 |
| Celeba (128x128) | 63.28 | 32.27 | 14.87 | 4.06 | 1.75 |

Table 2: The inference time (in seconds) of our IFM with different batch sizes $|B|$ in generation.

### 4.3 IMAGE-TO-IMAGE TRANSLATION

Following (Zhu et al., 2017; Kolesov et al., 2025), we also consider an unpaired image translation task with two scenarios: translating $32 \times 32$ colored MNIST digits from '2' to '3' (2→3) and translating $64 \times 64$ scenes from Winter to Summer (W→S). The placement of images on the hyperplanes follows the same setup as the §4.2. We learn the normalized field between the plates using both independent and mini-batch optimal transport plans. Our IFM and IFM-MB approaches effectively preserve shapes (with IFM-MB performing slightly better due to the use of the transport plan) and changes the styles of the initial images (see Figs. 10a and 10b). For completeness, we compare IFM and IFM-MB with popular image-to-image translation methods, including Flow Matching, diffusion-based (Ho et al., 2020, DDIB), adversarial (Zhu et al., 2017, CycleGAN), and EFM. We

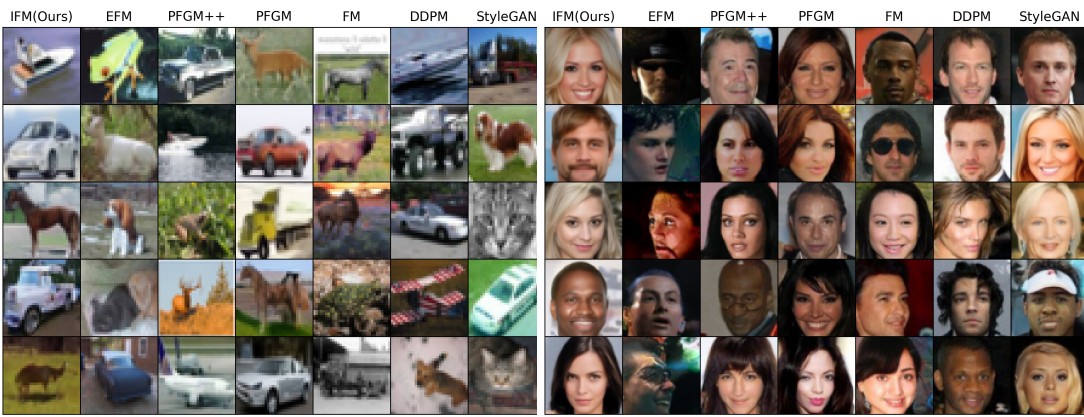

(a) CIFAR-10 32x32.                    (b) Celeba 64x64.

Figure 9: *Image Generation:* Samples obtained by **IFM**(ours) with the independent plan, electrostatic-based approaches **EFM** and **PFGM&PFGM++**, flow-based **FM**, diffusion-based **DDPM** and **StyleGAN**.

evaluate the performance of these methods using the CMMD Yan et al. (2022), as reported in Table 3. We also add a discussion of the distinctions between our approach and FM in the Appendix F .

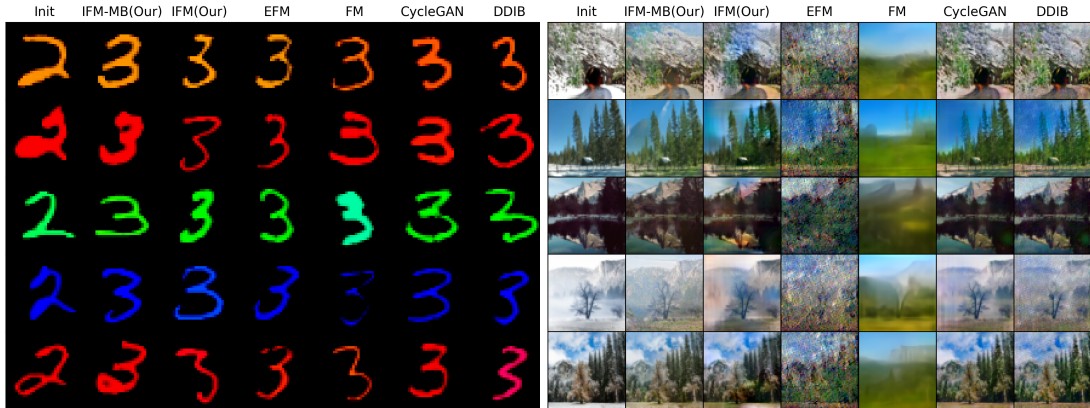

(a) Colored digits '2' → Colored digits '3'.                    (b) Winter → Summer.

Figure 10: *Image Translation:* Samples obtained by **IFM**(ours) with/without the minibatch plan, electrostatic-based approach **EFM**, flow-based **FM**, diffusion-based **DDIB** and adversarial **CycleGAN**.

| Dataset / Method | IFM-MB (our) | IFM (our) | EFM | FM | CycleGAN | DDIB |
|---|---|---|---|---|---|---|
| '2' → '3' (32x32) | 0.87 | 0.95 | 0.93 | 1.06 | 0.90 | 0.96 |
| W→S (64x64) | 1.13 | 1.25 | ≫1 | ≫1 | 1.33 | 1.39 |

Table 3: *Unpaired Image Translation:* CMMD↓ on W→S and colored difits '2'→'3' for our **IFM**, **EFM**, **FM**, **CycleGAN** and **DDIB**.

**Computational efficiency** of our method in translation is almost the same as in generation (§4.2). Also, the core implementation components of IFM (artitectures, number of steps in ODE solver, etc.) are the same as in generation and are again identical to the closest ODE-based competitors EFM & FM to ensure the fair comparison. Therefore, the runtime & memory usage is the same for all these mentioned approaches. For completeness, we report the inference time of IFM in Table 4 below.

| Dataset / Batch Size | 256 | 128 | 64 | 16 | 1 |
|---|---|---|---|---|---|
| MNIST '2'→'3' (32x32) | 10.95 | 5.73 | 1.61 | 0.85 | 0.73 |
| W → S (64x64) | 36.84 | 18.44 | 8.53 | 4.05 | 1.76 |

Table 4: The inference time (in seconds) of our IFM with different batch sizes $|B|$ in translation.

## 5 DISCUSSION

Our proposed IFM method is a generalization of EFM allowing using rather general interaction fields for distribution transfer. Our implementation of the particular IFM field is just one of many possible realizations. The search for a more optimal realization is a difficult task and is a promising subject of future research that opens opportunities for the further development of electrostatic-inspired models.

Our IFM overcomes major limitations of prior EFM method (§2.3): backward-oriented field lines, line termination problems, and training volume selection issues. Moreover, due to our field's specific structure, we also address the high-dimensionality challenge and the associated numerical instability that affects EFM and PFGM due to the Coulomb factor $1/\|\widetilde{\mathbf{x}} - \widetilde{\mathbf{x}}'\|^D$ (see Appendix C).

**Impact statement.** Our paper presents work with a goal to advance ML. There are many potential societal consequences of our work, none of which we feel must be specifically highlighted here.

**Reproducibility.** We provide the experimental details in Appendix B and the code to reproduce the conducted experiments in the supplementary materials (see README.md).

**Acknowledgment.** The work was supported by the grant for research centers in the field of AI provided by the Ministry of Economic Development of the Russian Federation in accordance with the agreement 000000C313925P4F0002 and the agreement №139-10-2025-033. We thank Kirill Sokolov for providing feedback and suggestions for improving the proofs and clarity of our paper.

**LLM Usage.** Large Language Models (LLMs) were used only to assist with rephrasing sentences and improving the clarity of the text. All scientific content, results, and interpretations in this paper were developed solely by the authors.

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

## A INTERACTION FIELD MATCHING: PROPERTIES AND PROOFS

### A.1 PROPERTIES OF INTERACTION FIELD LINES

In this section we formulate several properties of the interaction field which are essential for the proof of the main theorem. First of all, let us formulate the notion of field flux which has been introduced intuitively in the main text.

**Definition A.1.** (Field flux). Consider an element of area $\mathbf{dS}$. The field flux $\mathbf{E}$ through this element is called $d\Phi = \mathbf{E} \cdot \mathbf{dS}$. If we need to calculate the field flux through a finite surface $\Sigma$, then the field flux is

$$\Phi = \int_\Sigma d\Phi = \int_\Sigma \mathbf{E} \cdot \mathbf{dS}. \tag{13}$$

Intuitively, field flux indicates how many field lines pass through a surface $\Sigma$. The greater the flux, the higher the number of lines passing through a given area.

**Remark**. For the closed surfaces, we assume that the normal is always directed outward.

**Lemma A.1** (Generalized Gauss theorem). *Let $\mathbb{P}$ and $\mathbb{Q}$ be two probability distributions on $\mathbb{R}^D$, and let $supp(\mathbb{P})$ and $supp(\mathbb{Q})$ denote their respective compact supports. These supports are located on the planes $z = 0$ and $z = L$ in the extended space $\mathbb{R}^{D+1}$, and the distributions satisfy $\int_{supp(\mathbb{P})} \mathbb{P}(\mathbf{x}_q)d\mathbf{x}_q = \int_{supp(\mathbb{Q})} \mathbb{Q}(\mathbf{x}_{\bar{q}})d\mathbf{x}_{\bar{q}}$. Let $\mathbf{E}_{q\bar{q}}(\widetilde{\mathbf{x}}) \equiv \mathbf{E}_{\mathbf{x}_q \mathbf{x}_{\bar{q}}}(\widetilde{\mathbf{x}})$ be an interaction field produced by a pair of a unit quark and a unit antiquark satisfying properties 1-3 of §3.2 with the transport plan $\pi(\mathbf{x}_q, \mathbf{x}_{\bar{q}})$. Let $M \subset \mathbb{R}^{D+1}$ be an open bounded set and let $\partial M$ be its boundary (a piecewise smooth hypersurface) such that $M \cap supp(\mathbb{P}) \neq \varnothing$ and $M \cap supp(\mathbb{Q}) = \varnothing$. Then*

$$\iint_{\partial M} \mathbf{E} \cdot \mathbf{dS} = \Phi_0 \cdot \int_M \mathbb{P}(\widetilde{\mathbf{x}})d\widetilde{\mathbf{x}}, \tag{14}$$

*where $\Phi_0 = \iint_{\partial M} \mathbf{E}_{q\bar{q}} \cdot \mathbf{dS}$ is the field flux from a single unit $q\bar{q}$-pair[*], and $\mathbb{P}(\widetilde{\mathbf{x}}) = \mathbb{P}(\mathbf{x})\delta(z)$.*

*Proof.* Substituting the explicit expression (7) for the interaction field $\mathbf{E}(\widetilde{\mathbf{x}})$ we obtain:

$$\iint_{\partial M} \mathbf{E} \cdot \mathbf{dS} = \iint_{\partial M} \left( \int_S \mathbf{E}_{q\bar{q}} \pi(\mathbf{x}_q, \mathbf{x}_{\bar{q}})d\mathbf{x}_q d\mathbf{x}_{\bar{q}} \right) \cdot \mathbf{dS} =$$

$$= \int_S \pi(\mathbf{x}_q, \mathbf{x}_{\bar{q}})d\mathbf{x}_q d\mathbf{x}_{\bar{q}} \cdot \iint_{\partial M} \mathbf{E}_{q\bar{q}} \cdot \mathbf{dS} =$$

$$= \int_S \pi(\mathbf{x}_q, \mathbf{x}_{\bar{q}})d\mathbf{x}_q d\mathbf{x}_{\bar{q}} \cdot \Phi_0 \Theta\left[\mathbf{x}_q \in supp(\mathbb{P}) \cap M\right] = \tag{15}$$

$$= \Phi_0 \int_{supp(\mathbb{P}) \cap M} d\mathbf{x}_q \int_{supp(\mathbb{Q})} d\mathbf{x}_{\bar{q}} \pi(\mathbf{x}_q, \mathbf{x}_{\bar{q}}) =$$

$$= \Phi_0 \cdot \int_M \mathbb{P}(\widetilde{\mathbf{x}})d\widetilde{\mathbf{x}}.$$

where $S = supp(\mathbb{P}) \times supp(\mathbb{Q})$ denotes the Cartesian product of the supports, and $\Theta[\mathbf{x}_q \in supp(\mathbb{P}) \cap M]$ is the indicator function equal to 1 if $\mathbf{x}_q \in supp(\mathbb{P}) \cap M$, and 0 otherwise.

The chain of equalities proceeds as follows: first, we substitute the definition of $\mathbf{E}(\widetilde{\mathbf{x}})$ from equation 7; then we interchange the order of integration (justified by the continuity of $\pi$ and $\mathbf{E}q\bar{q}$ and the compactness of their supports); next, we use that the flux of a single pair $\mathbf{E}q\bar{q}$ through $\partial M$ equals $\Phi_0$

---

[*]We assume (§3.2) that the flux $\Phi_{q\bar{q}}$ is proportional to the charge of the quark $q$ that creates field $\mathbf{E}_{q\bar{q}}$ and does not depend on the relative position of the quark-antiquark pair. That is, all other things being equal, the replacement e.g. $q \to 2q$ will lead to $\Phi_{q\bar{q}} \to 2\Phi_{q\bar{q}}$. Therefore, for any pair $q\bar{q}$ with the same unit charge $q = 1$ (which create an elementary field $\mathbf{E}_{q\bar{q}}$), the flux $\Phi_{q\bar{q}} \equiv \Phi_{1\bar{1}} = \Phi_0$ will be the same for all pairs

precisely when the quark lies inside $M$, which is expressed by the indicator; finally, we integrate over $\mathbf{x}_{\bar{q}}$ to obtain the required result $\Phi_0 \cdot \int_M \mathbb{P}(\widetilde{\mathbf{x}})d\widetilde{\mathbf{x}}$

$\square$

**Remark**. If $M$ contains a part of the distribution $\mathbb{Q}(\cdot)$ but does not contain $\mathbb{P}(\cdot)$, then the statement of the theorem will be written as follows:

$$\iint\limits_{\partial M} \mathbf{E} \cdot \mathbf{dS} = \Phi_0 \cdot \int_M \mathbb{Q}(\widetilde{\mathbf{x}})d\widetilde{\mathbf{x}}. \tag{16}$$

**Corollary A.2.** *For any point $\widetilde{\mathbf{x}}$ in the support of distribution $\mathbb{P}(\cdot)$:*

$$E_z^+(\widetilde{\boldsymbol{x}}) - E_z^-(\widetilde{\boldsymbol{x}}) = \Phi_0 \cdot \mathbb{P}(\boldsymbol{x}). \tag{17}$$

*where $E_z^{\pm}(\widetilde{\boldsymbol{x}}) = E_z(\boldsymbol{x}, 0^{\pm})$.*

*Proof.* Consider a volume $\mathbf{dS} \in \text{supp } \mathbb{P}(\cdot)$. Consider a closed surface, a cylinder with infinitesimal indentation in different directions in the plane $z = 0$, see Fig. 11.

The flux through this surface consists of three summands: the $d\Phi^+$ flux in the positive direction of the $z$-axis, the $d\Phi^-$ flux in the negative direction of the $z$-axis, and the $d\Phi_{\text{lat}}$ flux through the lateral surface:

$$d\Phi_{\text{full}} = d\Phi^+ + d\Phi^- + d\Phi_{\text{lat}} = E_z^+ dS - E_z^- dS + 0. \tag{18}$$

Here $d\Phi_{\text{lat}} = 0$ since the height of the cylinder under consideration can be made as small as we want (infinitesimal of higher order than $dS$). $d\Phi^- = -E_z^- dS$ has a negative sign due to the fact that the normal to the closed surface is directed outward, i.e. in the opposite direction from the axis $z$.

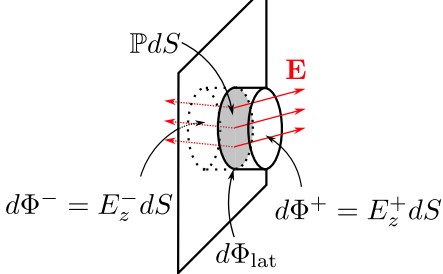

Then, due to the generalized Gauss's theorem:

$$d\Phi_{\text{full}} = (E_z^+ - E_z^-)dS = \Phi_0 \cdot \mathbb{P}dS \tag{19}$$

which proves the corollary.

$d\Phi^- = E_z^- dS \qquad d\Phi^+ = E_z^+ dS$

$d\Phi_{\text{lat}}$

Figure 11: Considered area.

**Lemma A.3** (On field lines). *Let $\mathbb{P}(\cdot)$ and $\mathbb{Q}(\cdot)$ be two (discrete or continuous) distributions corresponding to quarks and antiquarks. Let these distributions have compact support and satisfy $\int \mathbb{P}(\mathbf{x}_q)d\mathbf{x}_q = \int \mathbb{Q}(\mathbf{x}_{\bar{q}})d\mathbf{x}_{\bar{q}}$. Let the field of the unit quark-antiquark pair $\mathbf{E}_{\mathbf{x}_q,\mathbf{x}_{\bar{q}}}(\widetilde{\mathbf{x}}) \equiv \mathbf{E}_{q\bar{q}}(\widetilde{\mathbf{x}})$ start at $\mathbf{x}_q$ and end at $\mathbf{x}_{\bar{q}}$ (Property 1), and conserve flux along the current tube (Property 2). Then the total field (7) from all quarks and antiquarks satisfies:*

*(a) Its lines start at $\mathbb{P}(\cdot)$ and end at $\mathbb{Q}(\cdot)$, except perhaps for the number of lines of zero flux*

*(b) It conserves flux along the current tubes.*

*Proof.* Let us begin with the proof of the second property. Let the field $\mathbf{E} = \int \mathbf{E}_{q\bar{q}} \pi_{q\bar{q}} d\mathbf{x}_q d\mathbf{x}_{\bar{q}}$ and $\int \mathbf{E}_{q\bar{q}} \cdot \mathbf{dS} = \text{const}$. Then:

$$\int \mathbf{E} \cdot \mathbf{dS} = \int \int \pi_{q\bar{q}} d\mathbf{x}_q d\mathbf{x}_{\bar{q}} \int \mathbf{E}_{q\bar{q}} \cdot \mathbf{dS} = \int \pi_{q\bar{q}} d\mathbf{x}_q d\mathbf{x}_{\bar{q}} \cdot \text{const} = \text{const} \cdot 1 = \text{const}. \tag{20}$$

Let us now prove the first property. Suppose the opposite. Let there be lines starting at $\mathbb{P}$ but not ending at $\mathbb{Q}$ (the case of lines not starting at $\mathbb{P}$ but ending at $\mathbb{Q}$ can be considered similarly).

Consider a $q\bar{q}$-pair. The field $\mathbf{E}_{q\bar{q}}$ of this pair by assumption of the lemma starts on $q$ and ends on $\bar{q}$, is continuous, and therefore cannot go to infinity. Therefore, the flux through any $\Sigma$-area infinitely distant from $q\bar{q}$-pair must be zero [†]:

$$\Phi_{q\bar{q}}^{\Sigma} = \int_{\Sigma} \mathbf{E}_{q\bar{q}} \cdot \mathbf{dS} \to 0, \|\mathbf{x}\| \to \infty. \tag{21}$$

Then, if $\mathbb{P}(\cdot)$ and $\mathbb{Q}(\cdot)$ have a compact support, the field flux from all quarks will also be zero:

$$\Phi_{\infty} = \int_{\Sigma} \mathbf{E} \cdot \mathbf{dS} = \int_{\Sigma} \int \mathbf{E}_{q\bar{q}} \pi_{q\bar{q}} d\mathbf{x}_q d\mathbf{x}_{\bar{q}} \cdot \mathbf{dS} = \int \pi_{q\bar{q}} \Phi_{q\bar{q}}^{\Sigma} d\mathbf{x}_q d\mathbf{x}_{\bar{q}} \to 0, \|\mathbf{x}\| \to \infty. \tag{22}$$

Therefore, if the lines under consideration go to infinity, their flux is zero.

Let us now consider lines ending not at infinity and not at the target distribution (i.e., at some intermediate points)[‡]. At the stopping point $\mathbf{E} = 0$ (otherwise it is not a stopping point). But then the total flux produced by such lines:

$$\Phi = \int \mathbf{E} \cdot \mathbf{dS} = 0 \tag{23}$$

which proves the statement.

$\square$

## A.2 Definition of stochastic map $T$

Movement from one distribution to the other is carried out along the field strength lines. In this section, we rigorously define this movement using a stochastic map $T$, taking into account the existence of two series of lines - forward-oriented and backward-oriented.

We define the *stochastic* **forward map** $T_F$ from $\mathrm{supp}(\mathbb{P})$ to $\mathrm{supp}(\mathbb{Q})$ through forward-oriented field lines. For this, we consider a point $\widetilde{\mathbf{x}}_q = (\mathbf{x}_q, \varepsilon), \varepsilon \to 0^+$ slightly shifted in the direction of the second plate. Let us move along the corresponding field line by integrating $d\widetilde{\mathbf{x}}(t) = \mathbf{E}(\widetilde{\mathbf{x}}(t))dt$.

Sooner or later, such movement leads to the intersection of the plane $z = 0$ or $z = L$. At this moment, the question arises of whether to continue or stop the movement. If the movement continues, it will proceed until the plane $z = 0$ or $z = L$ is intersected again. Then, a decision must again be made - to stop or to go further. This procedure must be continued until we reach the final stopping point. Let us denote the intersection points as follows:

$$\widetilde{\mathbf{x}}_q \to \widetilde{\mathbf{x}}_F^{(1)} \to \widetilde{\mathbf{x}}_F^{(2)} \to ... \to \widetilde{\mathbf{x}}_F^{(N)} \tag{24}$$

At each of these points, it is necessary to stop with probability $\nu(\widetilde{\mathbf{x}}_F^{(i)})$ and continue movement with probability $1 - \nu(\widetilde{\mathbf{x}}_F^{(i)})$, where

$$\nu(\widetilde{\mathbf{x}}_F^{(i)}) = \begin{cases} 1, & \text{if } E_z^{(\text{before})}(\widetilde{\mathbf{x}}_F^{(i)}) \text{ and } E_z^{(\text{after})}(\widetilde{\mathbf{x}}_F^{(i)}) \text{ have opposite signs} \\ 0, & \text{if } |E_z^{(\text{after})}(\widetilde{\mathbf{x}}_F^{(i)})| \geq |E_z^{(\text{before})}(\widetilde{\mathbf{x}}_F^{(i)})| \\ \frac{|E_z^{(\text{before})}(\widetilde{\mathbf{x}}_F^{(i)})| - |E_z^{(\text{after})}(\widetilde{\mathbf{x}}_F^{(i)})|}{|E_z^{(\text{before})}(\widetilde{\mathbf{x}}_F^{(i)})|} & \text{if } |E_z^{(\text{after})}(\widetilde{\mathbf{x}}_F^{(i)})| < |E_z^{(\text{before})}(\widetilde{\mathbf{x}}_F^{(i)})| \end{cases}, \tag{25}$$

---

[†]The difference $\Phi_{q\bar{q}}^{\Sigma}$ with the $\Phi_0$ here is that $\Sigma$ is an open surface infinitely distant from the $q\bar{q}$-pair, while $\Phi_0$ is the flux through a closed surface $\partial M$ containing charge $q$ and not containing charge $\bar{q}$. In other words, $\Phi_0$ denotes the total field flux between a quark-antiquark pair of unit charge, while $\Phi_{q\bar{q}}^{\Sigma}$ defines the field flux through some infinitely distant surface.

[‡]Note that there are only two possible options - either the field line goes to infinity, or does not go to infinity, that is, it stops somewhere at an intermediate point.

where $E_z^{(\text{before})}(\widetilde{\mathbf{x}}_F^{(i)})$ and $E_z^{(\text{after})}(\widetilde{\mathbf{x}}_F^{(i)})$ are the values of the $z$-component of the field immediately before and immediately after intersecting the plane at point $\widetilde{\mathbf{x}}_F^{(i)}$, respectively.

To understand the meaning of this probability, note that if $E_z^{(\text{before})}(\widetilde{\mathbf{x}}_F^{(i)})$ and $E_z^{(\text{after})}(\widetilde{\mathbf{x}}_F^{(i)})$ have opposite signs, then further movement along the field lines is impossible (and thus we have arrived at the final point $\widetilde{\mathbf{x}}_F^{(N)}$ on the target distribution). If they have the same sign, then further movement along the field lines is possible.

If further movement is possible, two situations arise: either $|E_z^{(\text{after})}(\widetilde{\mathbf{x}}_F^{(i)})| \geq |E_z^{(\text{before})}(\widetilde{\mathbf{x}}_F^{(i)})|$ or $|E_z^{(\text{after})}(\widetilde{\mathbf{x}}_F^{(i)})| < |E_z^{(\text{before})}(\widetilde{\mathbf{x}}_F^{(i)})|$. In the first case, after crossing the plane, the field magnitude (and consequently the flux) increases - therefore the stopping probability is set to zero. If the flux magnitude becomes smaller after crossing, it means that one must stop with some probability and continue with another. Lemma X explains why this particular probability value is chosen for the latter situation.

The *stochastic* **backward map** $T_B$ is constructed similarly using left limit $\varepsilon \to 0^-$ and backward-oriented field lines.

The complete transport $T$ is then described by the *random variable*:

$$T(\mathbf{x}_q) = \begin{cases} T_F(\mathbf{x}_q) \text{ with probability } \mu(\mathbf{x}_q), \\ T_B(\mathbf{x}_q) \text{ with probability } 1 - \mu(\mathbf{x}_q), \end{cases} \tag{26}$$

capturing both forward and backward trajectory endpoints for each $\mathbf{x}_q \in \text{supp}(\mathbb{P})$ with probabilities $\mu(\mathbf{x}_q)$ and $1 - \mu(\mathbf{x}_q)$, where

$$\mu(\mathbf{x}_q) = \begin{cases} 0, & E_z^+ < 0 \\ 1, & E_z^- > 0 \\ \frac{E_z^+}{E_z^+ + |E_z^-|} & \text{otherwise.} \end{cases} \tag{27}$$

Here $E_z^{\pm} = E_z(\widetilde{\mathbf{x}} \pm \varepsilon \mathbf{e}_z)$, $\varepsilon \to 0^+$ are the left and right limits of the field value at the point $\widetilde{\mathbf{x}}$ in $z$ direction. The meaning of this probability is as follows. Value $\mu(\mathbf{x}_q)$ allows one to choose a forward or backward sets of lines with a probability proportional to the field flux in the corresponding direction (i.e., proportional to $E_z^+$ and $|E_z^-|$, respectively). At the same time, if it is impossible to move forward ($E_z^+ < 0$), the map $T_B$ is chosen ($\mu(\mathbf{x}_q) = 0$), and if it is impossible to move backward ($E_z^- > 0$), the map $T_F$ is chosen ($\mu(\mathbf{x}_q) = 1$).

## A.3 IFM MAIN THEOREM PROOF

**Lemma A.4** (First lemma on the flow). *Let $\mathbb{P}(\cdot)$, $\mathbb{Q}(\cdot)$ be two D-dimensional data distributions having a compact support, located on the planes $z = 0$ and $z = L$ in $\mathbb{R}^{D+1}$, respectively. Let $\{\widetilde{\mathbf{x}}_{q_i}\}_{i=1}^n$ be a sample of points distributed over $\mathbb{P}$. Let $dS$ be an element of D-dimensional area on the distribution of $\mathbb{P}$ ($dS \in \text{supp } \mathbb{P}$). Let the field near the element $dS$ have different signs: $E_z^+ > 0$ and $E_z^- < 0$. Let $dn$ be the number of points from the sample that fall in the volume $dS$. Let $dn = dn_F + dn_B$, where $dn_F$ is the number of points from $dS$ that correspond to the mapping $T_F$ (i.e., movement along forward-oriented lines), and $dn_B$ corresponds to $T_B$. Then:*

$$\begin{aligned} \frac{dn_F}{n} &\xrightarrow[n\to\infty]{a.s.} \frac{E_z^+ dS}{\Phi_0} = \frac{d\Phi_F}{\Phi_0}, \\ \frac{dn_B}{n} &\xrightarrow[n\to\infty]{a.s.} \frac{|E_z^-| dS}{\Phi_0} = \frac{d\Phi_B}{\Phi_0}, \end{aligned} \tag{28}$$

*where $\left(\dots \xrightarrow[n\to\infty]{a.s.} \dots\right)$ denotes the almost sure convergence.*

*Proof.* According to the multiplication rule of probability and the law of large numbers:

$$\begin{aligned} \frac{dn_F}{n} &\xrightarrow[n\to\infty]{a.s.} (\text{probability of choosing } T_F) \cdot (\text{probability of falling in } dS) = \\ &= \mu(\widetilde{\mathbf{x}}) \cdot \mathbb{P}(\widetilde{\mathbf{x}}) dS = \frac{E_z^+}{E_z^+ + |E_z^-|} \cdot \frac{(E_z^+ + |E_z^-|) dS}{\Phi_0} = \frac{E_z^+ dS}{\Phi_0} = \frac{d\Phi_F}{\Phi_0}. \end{aligned} \tag{29}$$

In the second equality, the definitions of probability $\mu(\cdot)$, see Eq. (27), and Corollary A.2 were used. The case $dn_B$ is proved similarly. □

**Lemma A.5** (Second lemma on the flow). *Let* $\mathbb{P}, \mathbb{Q}, \{\widetilde{\boldsymbol{x}}_{q_i}\}_{i=1}^n, dS, dn$ *have the same meaning as in the Lemma A.4. Let* $E_z^+$ *and* $E_z^-$ *have the same sign near* $dS$ *(i.e., either simultaneously* $E_z^\pm > 0$ *or simultaneously* $E_z^\pm < 0$*). Then*

$$\frac{dn}{n} \xrightarrow[n\to\infty]{a.s.} \frac{d\Phi_{after}}{\Phi_0} - \frac{d\Phi_{before}}{\Phi_0}, \tag{30}$$

*where* $d\Phi_{before}$ *is the field flux through the current tube supported on* $dS$ *immediately before crossing the plane* $dS \in \operatorname{supp} \mathbb{P}(\cdot)$*, and* $d\Phi_{after}$ *is the flux after crossing.*

**Remark.** This statement implies that when the field crosses the plane $\mathbb{P}$ containing a charge (proportional to $dn/n$), the field flux must *increase* by $\Phi_0 \cdot dn/n$.

*Proof.* For clarity, consider the case $E_z^+ > 0$, $E_z^- > 0$ when $\mu(\mathbf{x}_q) = 1$ and motion processes only along the forward-oriented lines, corresponding to the mapping $T_F$.

By the probability product theorem, the strong law of large numbers, Corollary A.2, and the definition of flux:

$$\frac{dn}{n} \to \mu(\mathbf{x})\mathbb{P}(\mathbf{x})dS = 1 \cdot \mathbb{P}(\mathbf{x})dS = \frac{E_z^+ - E_z^-}{\Phi_0}dS = \frac{d\Phi_{after}}{\Phi_0} - \frac{d\Phi_{before}}{\Phi_0}. \tag{31}$$

□

Lemmas A.4 and A.5 address the behavior near the distribution $\mathbb{P}$. Similar statements are valid for the behavior near $\mathbb{Q}$. When moving along field lines, we eventually reach the plane $z = L$. At this point two different scenarios may occur:

1. $E_z^+(L)$ and $E_z^-(L)$ have opposite signs. Then the field line motion terminates in this case.
2. $E_z^+(L)$ and $E_z^-(L)$ have the same sign. Then a portion $dn'$ of lines must terminate, while others continue.

This portion $dn'$ can be found from the line termination property in $\mathbb{Q}$.

**Lemma A.6** (Line Termination). *If* $E_z^+(L)$ *and* $E_z^-(L)$ *have the same sign upon crossing* $z = L$*, the number of lines terminating on* $z = L$ *satisfies:*

$$\frac{dn'}{n} \to -\frac{d\Phi_{after}}{\Phi_0} + \frac{d\Phi_{before}}{\Phi_0}, \tag{32}$$

**Remark.** When the field crosses the plane $\mathbb{Q}$ containing a charge (proportional to $dn'/n$), the field flux must *decrease* by $\Phi_0 \cdot dn'/n$.

*Proof.* . Consider the current tube before it intersects the plane $z = L$. Let us denote the number of lines inside $dn_{before}$. As a result of the intersection $z = L$, some of the lines $dn'$ stop moving, while some of the lines $dn_{after}$ continue moving. In view of the first Lemma A.4 on flow, as well as the conservation of flow inside the current tube (Property 2 in §3.2):

$$\frac{dn_{before}}{n} \xrightarrow[n\to\infty]{a.s.} d\Phi_{before}. \tag{33}$$

Then, by virtue of the law of large numbers and the fact that $dn_{before} = dn' + dn_{after}$, we have:

$$\frac{dn'}{n} \xrightarrow[n\to\infty]{a.s.} (\text{probability of termination}) \cdot \frac{dn_{before}}{n} \xrightarrow[n\to\infty]{a.s.} \nu(\mathbf{x}^-) \cdot d\Phi_{before}$$
$$\frac{E_z^- - E_z^+}{E_z^-} \cdot E_z^- dS' = (E_z^- - E_z^+)dS' = -d\Phi_{after} + d\Phi_{before} \tag{34}$$

$\square$

We now proceed to prove the main theorem.

**Theorem A.7** (Interaction Field Matching). *Let $\mathbb{P}(\boldsymbol{x}_q)$ and $\mathbb{Q}(\boldsymbol{x}_{\bar{q}})$ be two data distributions that have compact support. Let $\boldsymbol{x}_q$ be distributed over $\mathbb{P}(\boldsymbol{x}_q)$. Then $\boldsymbol{x}_{\bar{q}} = T(\boldsymbol{x}_q)$ is distributed over $\mathbb{Q}(\boldsymbol{x}_{\bar{q}})$ almost surely:*

$$\text{If } \boldsymbol{x}_q \sim \mathbb{P}(\boldsymbol{x}_q) \Rightarrow T(\boldsymbol{x}_q) = \boldsymbol{x}_{\bar{q}} \sim \mathbb{Q}(\boldsymbol{x}_{\bar{q}}). \tag{35}$$

*Proof.* Let $\{\mathbf{x}_{q_i}\}_{i=1}^n$ be points distributed according to $\mathbb{P}$. Moving along the field lines via mapping $T$, we obtain points $\mathbf{x}_{\bar{q}_i} = T(\mathbf{x}_{q_i})$ in distribution $\mathbb{Q}$.

Consider a $D$-dimensional area element $dS' \subset \operatorname{supp}\mathbb{Q}$. Let $dn'$ be the number of points $\mathbf{x}_{\bar{q}_i}$ in this area. Define the sample:

$$\hat{\mathbb{Q}}_n dS' = \frac{dn'}{n}. \tag{36}$$

The aim is to prove

$$\hat{\mathbb{Q}}_n \to \mathbb{Q}. \tag{37}$$

The points $dn'$ arrive via forward or backward directions:

$$dn' = dn'_F + dn'_B. \tag{38}$$

Consider $dn'_F$ and its associated flux tube. Traverse this tube inversely along the field lines until stopping at $\mathbb{P}$. During this motion, multiple crossings of $z = 0$ and/or $z = L$ may occur. Denote the intersection points:

$$\mathbf{x}_{\bar{q}} = \mathbf{x}_0 \to \mathbf{x}_1 \to \cdots \to \mathbf{x}_{N-1} \to \mathbf{x}_N = \mathbf{x}_q. \tag{39}$$

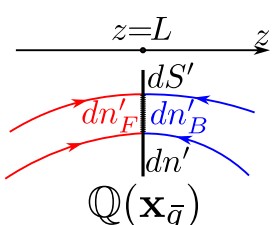

Figure 12: Points corresponding to forward and backward lines.

Their corresponding area elements are

$$dS' = dS_0 \to dS_1 \to \cdots \to dS_{N-1} \to dS_N = dS. \tag{40}$$

Point counts in these areas read

$$dn' = dn_0 \to dn_1 \to \cdots \to dn_{N-1} \to dn_N = dn, \tag{41}$$

where $dn_k$ $(k = 0, ..., N+1)$ is number of points from sample $\{\mathbf{x}_{q_i}\}_{i=1}^n$ or from map $\{T(\mathbf{x}_{q_i})\}_{i=1}^n$ inside the volume $dS_k$ near point $\mathbf{x}_k$ that corresponds to considered motion inside current tube.

The $dn_k$ are not arbitrary but related by flux conservation. Only the charged planes ($z = 0$ or $z = L$) can alter the count:

- At $z_i = 0$: Line count increases by $dn_i$
- At $z_i = L$: Line count decreases by $dn_i$

Mathematically:

$$\sum_{i=0}^{N} (-1)^{f_i} dn_i = 0, \tag{42}$$

where

$$f_i = \begin{cases} 0 & \text{if } z_i = 0, \\ 1 & \text{if } z_i = L. \end{cases} \tag{43}$$

Due to the first Lemma on flow A.4:

$$\frac{dn_N}{n} \equiv \frac{dn}{n} \xrightarrow[n\to\infty]{\text{a.s.}} \frac{d\Phi_N}{\Phi_0} \equiv \frac{d\Phi}{\Phi_0}, \tag{44}$$

Due to the second Lemma A.5 on the flow , and because of the line termination Lemma A.3:

$$(-1)^{f_i} \cdot \frac{dn_i}{n} \xrightarrow[n\to\infty]{\text{a.s.}} \frac{d\Phi_{\text{after},i}}{\Phi_0} - \frac{d\Phi_{\text{before},i}}{\Phi_0}. \tag{45}$$

According to the law of conservation of flux along the tube (Lemma A.3):

$$d\Phi_{\text{after},i} = d\Phi_{\text{before},i-1}. \tag{46}$$

Whence we obtain a chain of equalities:

$$0 = \sum_{k=0}^{N+1}(-1)^{f_k}dn_k = -\frac{dn'_F}{n} + (-1)^{f_1}\frac{dn_1}{n} + ... + (-1)^{f_N}\frac{dn_N}{n} + \frac{dn}{n} \Rightarrow$$

$$\frac{dn'_F}{n} = (-1)^{f_1}\frac{dn_1}{n} + ... + (-1)^{f_N}\frac{dn_N}{n} + \frac{dn}{n} \xrightarrow[n\to\infty]{\text{a.s.}} \tag{47}$$

$$\xrightarrow[n\to\infty]{\text{a.s.}} -d\Phi_{\text{after},1} - d\Phi_{\text{before},1} + ... + d\Phi_{\text{after},N} - d\Phi_{\text{before},N} + d\Phi_{N+1} =$$

$$= d\Phi_{\text{after},1} + 0 + ... + 0 = d\Phi'_F.$$

Consequently,

$$\frac{dn'_F}{n} \xrightarrow[n\to\infty]{\text{a.s.}} \frac{d\Phi'_F}{\Phi_0}. \tag{48}$$

Similarly, it can be proven that

$$\frac{dn'_B}{n} \xrightarrow[n\to\infty]{\text{a.s.}} \frac{d\Phi'_B}{\Phi_0}. \tag{49}$$

Then, by virtue of the generalized Gauss's theorem (Lemma A.1), we finally have

$$\hat{\mathbb{Q}}_n dS' = \frac{dn'}{n} = \frac{dn'_F}{n} + \frac{dn'_B}{n} \xrightarrow[n\to\infty]{\text{a.s.}} \frac{d\Phi'_F}{\Phi_0} + \frac{d\Phi'_B}{\Phi_0} = \mathbb{Q}dS. \tag{50}$$

This completes the proof.

### A.4 INTERACTION FIELD REALIZATION

Here we formulate an algorithm for computing our constructed field which is inpired by strong interaction in physics at an arbitrary point $\widetilde{\mathbf{x}} \in \mathbb{R}^{D+1}$ with the quark $q$ and the antiquark $\bar{q}$ being at $\widetilde{\mathbf{x}}_q$ and $\widetilde{\mathbf{x}}_{\bar{q}}$ (see Fig. 6).

**Symmetric case.** Let a quark $q$ be located at the origin: $\widetilde{\mathbf{x}}_q = (\mathbf{0},0) \in \mathbb{R}^{D+1}$, and the antiquark $\bar{q}$ at the point $\widetilde{\mathbf{x}}_{\bar{q}} = (\mathbf{0},L) \in \mathbb{R}^{D+1}$. The arbitrary point of space can be written as $\widetilde{\mathbf{x}} = (\mathbf{x}_\perp, z) = \widetilde{\mathbf{x}}_\perp + z\mathbf{e}_z$, where $\widetilde{\mathbf{x}}_\perp \in \mathbb{R}^{D+1}$ is the component of the vector $\widetilde{\mathbf{x}}$ orthogonal to the $z$-axis. We introduce the following string hyperparameters (Fig. 6a):

- $\sigma_0$ is the effective width of the string in the cross section.

- $d$ is the size of the region of the string in which the field lines will curve toward the quark (antiquark). Thus, in the interval $z \in [d, L-d]$ the field lines are straight, and in the regions $z \in [0, d]$ and $z \in [L-d, L]$ the lines will be curved. Value $k = \pi/2d$ is also introduced.

We define the dependence of the effective string width $\sigma(z)$ on the coordinate $z$ as follows:

---

**Algorithm 3** Interaction field calculation

---

**Input:** Positions of quark and antiquark: $\widetilde{\mathbf{x}}_q, \widetilde{\mathbf{x}}_{\bar{q}} \in \mathbb{R}^{D+1}$, with $z_q = 0, z_{\bar{q}} = L$
  Arbitrary point $\widetilde{\mathbf{x}} \in \mathbb{R}^{D+1}$
  String hyperparameters: $\sigma_0, d, k = \pi/2d$
**Output:** The interaction field $\mathbf{E}_{q\bar{q}}(\widetilde{\mathbf{x}})$
**Algorithm:**
  Calculate the vector connecting the quarks: $\widetilde{\mathbf{r}} = \widetilde{\mathbf{x}}_{\bar{q}} - \widetilde{\mathbf{x}}_q \in \mathbb{R}^{D+1}$
  Calculate the unit direction vector corresponding to it: $\mathbf{e}'_z = \frac{\widetilde{\mathbf{r}}}{||\widetilde{\mathbf{r}}||} \in \mathbb{R}^{D+1}$
  Calculate the vector of shift of the point $\widetilde{\mathbf{x}}$ from the axis of the string:

$$\widetilde{\rho} = \widetilde{\mathbf{x}} - \widetilde{\mathbf{x}}_q + (\widetilde{\mathbf{x}}_q - \widetilde{\mathbf{x}}_{\bar{q}})\frac{z}{L} \in \mathbb{R}^{D+1}$$

,
  where $z$ is corresponding coordinate of point $\widetilde{\mathbf{x}}$
  Calculate $x_\perp = ||\widetilde{\rho}||, \mathbf{e}_\perp = \frac{\widetilde{\rho}}{||\widetilde{\rho}||}$
  Calculate the string width $\sigma(z)$ according to (51)
  Calculate the angle $\alpha(x_\perp, z)$ according to (53)
  Calculate the value of field $E(x_\perp, z)$ according to (54)
  Calculate the direction $\mathbf{n}(\widetilde{\mathbf{x}}) = \cos\alpha(x_\perp, z) \cdot \mathbf{e}'_z + \sin\alpha(x_\perp, z) \cdot \mathbf{e}_\perp \in \mathbb{R}^{D+1}$
  Return: $\mathbf{E}_{q\bar{q}}(\widetilde{\mathbf{x}}) = E(x_\perp, z)\mathbf{n}(x_\perp, z)$

---

$$\sigma(z) = \begin{cases} \sigma_0 \sin(kz), & z \in [0, d], \\ \sigma_0, & z \in [d, L-d], \\ \sigma_0 \sin(k(L-z)), & z \in [L-d, d], \\ 0, & \text{otherwise.} \end{cases} \tag{51}$$

The field direction $\mathbf{n}(\widetilde{\mathbf{x}})$ at the point $\widetilde{\mathbf{x}}$ is defined as:

$$\mathbf{n}(\widetilde{\mathbf{x}}) = \cos\alpha(x_\perp, z) \cdot \mathbf{e}_z + \sin\alpha(x_\perp, z) \cdot \mathbf{e}_\perp \in \mathbb{R}^{D+1}, \tag{52}$$

where $\mathbf{e}_z, \mathbf{e}_\perp$ are the unit vectors along the $z$-axis and along the vector $\widetilde{\mathbf{x}}_\perp$, respectively, i.e., $\mathbf{e}_\perp = \widetilde{\mathbf{x}}_\perp/||\widetilde{\mathbf{x}}_\perp||$. $\alpha = \alpha(x_\perp, z)$ is the angle between the field direction at a given point and the $z$-axis. This angle is determined from the following considerations. Let $\widetilde{\mathbf{x}}'(z')$ be the field line parallel to the level $\sigma(z)$ (i.e. $\forall z' : x'_\perp(z')/\sigma(z') = \text{const}$) which passes through the point $(\mathbf{x}_\perp, z)$, i.e., $\widetilde{\mathbf{x}}'(z')|_{z'=z} = \widetilde{\mathbf{x}} = (\mathbf{x}_\perp, z)$. Then $\alpha = \alpha(x_\perp, z)$ is determined by $\tan\alpha = \frac{dx'_\perp}{dz'}\Big|_{z'=z}$:

$$\alpha = \alpha(x_\perp, z) = \begin{cases} \arctan(kx_\perp \cot(kz)), & z \in [0, d], \\ 0, & z \in [d, L-d], \\ \arctan(kx_\perp \cot(k(L-z))), & z \in [L-d, L]. \end{cases} \tag{53}$$

We define the field strength value as the product of the Gaussian distribution in the radial direction and a normalization factor that keeps the interaction field flux invariant along the tube:

$$E(x_\perp, z) = \exp\left(-\frac{x_\perp^2}{2\sigma(z)^2}\right) \cdot \frac{1}{\sigma(z)^D \cos\alpha(x_\perp, z)}. \tag{54}$$

**Shifted case**. In the case where the quarks are in the shifted positions $\widetilde{\mathbf{x}}_q$ and $\widetilde{\mathbf{x}}_{\bar{q}}$, we use a field shift parallel to the planes $z = 0$ and $z = L$, as shown in Fig. 6b. We use the shift and not the rotation of the string with the aim of not generating backward-oriented lines and lines traversing the region $z > L$. The detailed algorithm for calculating the field is formulated in Algorithm 3

### A.5 PROOF OF PROPERTIES OF INTERACTION FIELD REALIZATION

**Theorem A.8** (Properties of our interaction field realization). *Our realization of the interaction field $\mathbf{E}(\widetilde{\mathbf{x}})$ satisfies the fundamental Properties 1-2 in §3.2, with following additional characteristics:*

- **Z-Axis caging**: *Field lines never extend beyond $z > L$.*

- **Unidirectional Flow**: *No backward-oriented field lines exist.*

- **Centrosymmetrical arrangement**: $\mathbf{E}(\widetilde{\mathbf{x}}) = \mathbf{E}(r_\perp, z)$.

- **Radial Decay**: *Monotonic decrease in field strength away from axis:*

$$\frac{\partial \|\mathbf{E}\|}{\partial r_\perp} \leq 0 \quad with \quad \lim_{r_\perp \to \infty} \mathbf{E}(\widetilde{\mathbf{x}}) = \mathbf{0}.$$

- **Axial Alignment**: *Field becomes parallel to the string axis in middle region:*

$$\mathbf{E}(r_\perp, z) \parallel \mathbf{e}'_z \quad for \quad z \in [d, L - d].$$

*Proof.* The interaction field starts at a quark and ends at an antiquark (Property 1§3.2) due to the fact that the field direction $\mathbf{n}(\widetilde{\mathbf{x}}) = \cos\alpha(x_\perp, z) \cdot \mathbf{e}'_z + \sin\alpha(x_\perp, z) \cdot \mathbf{e}_\perp$ is a tangent to the curve $x'_\perp(z')$, which by construction begins at $\widetilde{\mathbf{x}}'_q$ and ends at $\widetilde{\mathbf{x}}_{\bar{q}}$.

Consider an infinitesimal current tube connecting a quark and an antiquark. The surface bounding this tube is parallel to the field line $x'_\perp(z')$. Along the field line by construction $x_\perp/\sigma(z) = \kappa = \text{const}$. In $(D+1)$-dimensional space, the area element $dS$ orthogonal to the $z$-axis is $dS \sim x_\perp^{D-1} dx_\perp$. Therefore, due to the definition of the flux and the explicit expression for $E(x_\perp, z)$ (54) we have

$$d\Phi = \mathbf{E} \cdot d\mathbf{S} = E dS \cos\alpha \sim \exp\left(-\frac{r_\perp^2}{2\sigma(z)^2}\right) \cdot \frac{1}{\sigma(z)^D \cos\alpha} \cdot x_\perp^{D-1} dx_\perp \cdot \cos\alpha =$$
$$= \exp\left(-\frac{\kappa^2}{2}\right) \cdot \kappa^{D-1} d\kappa = \text{const}.$$
(55)

Therefore, Property 2 §3.2 is satisfied.

The Z-Axis caging property is satisfied because $\sigma(z > L) = 0$. The Unidirectional Flow property is satisfied due to $\sigma(z < L) = 0$. The Cylindrical Symmetry property is satisfied because $\mathbf{E}(\widetilde{\mathbf{x}}) = \mathbf{E}(x_\perp, z)$. Radial decay property is satisfied because of the explicit formula (54) for $E(x_\perp, z)$. Finally, the Axial alignment property is satisfied because $\alpha(z \in [d, L - d]) = 0$. $\qquad\square$

**Remark.** A crucial element in the flux conservation proof is the factor $1/\sigma(z)^D$ in the definition of the IFM field (see (54)). The intuition behind this factor can be explained as follows: any flux tube must narrow to a point as it approaches a charge. Consequently, the cross-section (which is proportional to $\sigma(z)^D$) must also decrease. Flux conservation can only be maintained by a proportional increase in the field strength, see Figure 13 below.

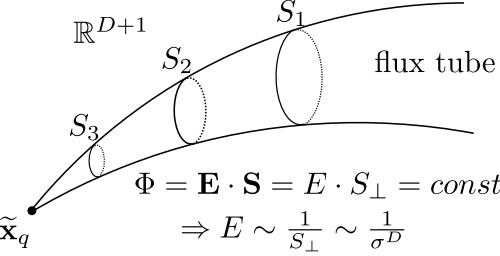

Figure 13: An illustration to the flux conservation property. To maintain the field flux conservation within a tube as it narrows, a proportional increase in field strength is required.

In simpler terms, the closer one is to a charge, the stronger the field must be.

## B EXPERIMENTAL DETAILS

We aggregate the hyper-parameters of our Algorithm 1 for different experiments in the Table 5. We base our code for the experiments on EFM's code `https://github.com/justkolesov/FieldMatching`. Our code is available at `https://github.com/justkolesov/InteractionFieldMatching`.

| Experiment | D | Batch Size | $L$ | $\sigma_0$ | $d$ | LR | $\pi$ plan |
|---|---|---|---|---|---|---|---|
| Gaussian Swiss-roll §4.1 | 2 | 1024 | [6, 40] | 1 | $[0.1, 0.5]L$ | 2e-4 | [Ind, MB] |
| CIFAR-10 Generation §4.2 | 3072 | 128 | 20 | 1 | $0.5L$ | 2e-4 | Ind |
| CelebA 64x64 Generation §4.2) | 12288 | 128 | 20 | 1 | $[0.1, 0.25, 0.4, 0.5]L$ | 2e-4 | Ind |
| MNIST digits 2→3 Translation §4.3 | 3072 | 128 | 20 | 1 | $0.5L$ | 2e-4 | [Ind, MB] |
| Winter→Summer Translation §4.3 | 12288 | 128 | 20 | 1 | $0.5L$ | 2e-4 | [Ind, MB] |
| CelebA 128x128 Generation App. D | 49152 | 128 | 20 | 1 | $0.5L$ | 2e-4 | Ind |
| Conditional CIFAR-10 generation App. E | 3072 | 128 | 20 | 1 | $0.5L$ | 2e-4 | Ind |

Table 5: Hyper-parameters of Alg. 1 for the experiments, where $D$ is the dimensionality of task, $L$ is the distance betwenn plates, $\sigma_0$ is the effective width, $d$ is the characteristic distance (see Fig.6a).

In the case of the Image experiments (see §4.3 and §4.2), we follow (Kolesov et al., 2025, EFM), (Xu et al., 2022; 2023, PFGM/PFGM++), (Ho et al., 2020, DDPM) and (Lipman et al., 2023, FM) and use Exponential Moving Averaging (EMA) technique with the ema rate decay equals 0.99 to provide smooth solution. Also, we use linear scheduler, that grows from 0 to $2e-4$ during the first 5000 iterations and decreases monotonically. As for the optimizer, we use Adam optimizer Kingma & Ba (2015) with the learning rate $2e-4$ and weight decay equals $1e-4$.

We compare our method with PFGM/PFGM++ Xu et al. (2022; 2023), whose the source code are taken from `https://github.com/Newbeeer/pfgmpp` for running **PFGM++** and `https://github.com/Newbeeer/Poisson_flow/` for **PFGM** in our experiments. We follow the proposed values of hyper parameters are appropriate for us: $\gamma = 5, \tau = 0.3, \epsilon = 1e-3$. The source code for **DDPM** is taken from `https://github.com/yang-song/score_sde_pytorch` with hyper-parameters $\sigma_{min} = 0.01, \sigma_{max} = 50, \beta_{min} = 0.1$ and $\beta_{max} = 20$. The source code for **FM** is taken from `https://github.com/facebookresearch/flow_matching` with linear interpolant . The source code for StyleGAN is taken from `https://github.com/NVlabs/stylegan2-ada-pytorch`.

## C ABLATION STUDY

Our IFM realization is defined by the following hyperparameters: the distance $L$ between plates, the string width $\sigma_0$, and the distance $d$ over which field lines curve toward the charges. In this Appendix, we address the practical selection of these hyperparameters and present an ablation study on how they affect our model's performance. We choose parameters based on the following ideas:

1. Since we learn the normalized field (see (11)), the factor $1/\sigma(z)^D$ cancels out. Indeed, let $\widetilde{\mathbf{x}} \in \mathbb{R}^{D+1}$ be a point where we estimate the normalized vector field using $B$ sampled pairs of quarks and anti-quarks. In accordance with the superposition principle (see (7)), the resulting field is obtained as the average of $B$ independent fields $E(x_\perp^{(i)}, z)\mathbf{n}(x_\perp^{(i)}, z)$ (see (54)) from each pair:

$$\frac{\mathbf{E}(\widetilde{\mathbf{x}})}{||\mathbf{E}(\widetilde{\mathbf{x}})||} = \sum_{i=1}^{B} \frac{\exp(-\frac{x_\perp^{(i)}}{2\sigma(z)^2})}{\sigma(z)^D}\mathbf{n}(x_\perp^{(i)}, z) \bigg/ \frac{1}{\sigma(z)^D}||\sum_{i=1}^{B}\exp(-\frac{x_\perp^{(i)}}{2\sigma(z)^2})\mathbf{n}(x_\perp^{(i)}, z)||.$$

Therefore, term $1/\sigma(z)^D$ cancels out completely. The practical choice of the hyperparameter $\sigma_0$ is determined solely by numerical considerations and is usually set to $\sigma_0 = 1$.

2. The distance $d$ should not be chosen too short—this complicates data translation via the ODE due to the high curvature of the field lines in the region $z \in [0, d] \cup [L-d, L]$ . In practice, we usually use $d \in [0.1L, \ 0.5L]$.

3. Finally, the distance $L$ does not significantly impact translation quality in our method (see Fig. 7). This is different from EFM, where making $L$ too large significantly worsens the results (see §2.3). Our IFM realization is specifically designed to reduce this dependency through straight field

segments for $z \in [d, L-d]$, where ODE integration follows straight lines. In practice, analogously to EFM, we set $L$ to be on the order of the data standard deviation: $L \sim \sqrt{D_{\mathbb{P}}}$ or $\sqrt{D_{\mathbb{Q}}}$.

Figure 14 presents a series of experiments with different values of the parameter $d$. It can be seen that the generation quality does not significantly depend on this parameter.

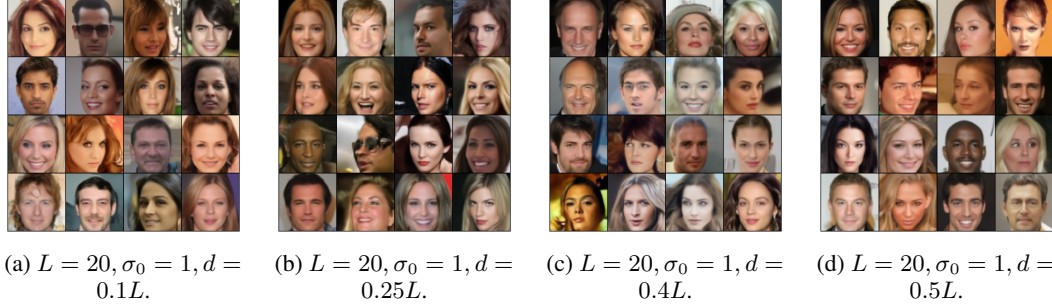

(a) $L = 20, \sigma_0 = 1, d = 0.1L$.

(b) $L = 20, \sigma_0 = 1, d = 0.25L$.

(c) $L = 20, \sigma_0 = 1, d = 0.4L$.

(d) $L = 20, \sigma_0 = 1, d = 0.5L$.

Figure 14: Image Generation on CelebA 64x64: Investigation of generation quality dependence on the model hyperparameter $d$ in our IFM method.

## D ADDITIONAL CELEBA GENERATION EXPERIMENT (128X128)

We also provide a more challenging image generation task on the 128×128 CelebA dataset. We follow the experimental design from the §4.2, placing the CelebA images and the noise from standard multivariate distribution $\mathcal{N}(0, I_{128 \times 128})$ on the left hyperplane ($z = 0$) and the right hyperplane ($z = 20$), respectively. We present the qualitative results of our IFM in Fig. 15, demonstrating its scalability in high-dimensional spaces.

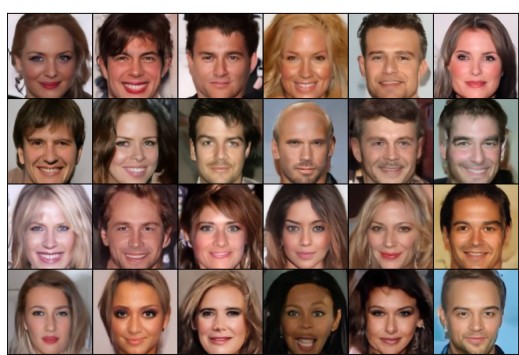

Figure 15: *Image Generation*: Samples obtained by IFM (ours) method with the independent transport plan on CelebA dataset with resolution 128x128.

## E CONDITIONAL IMAGE GENERATION ON CIFAR-10

Our IFM can be easily adapted to conditional generation tasks. For generating images of a specific class $c$, we learn a conditional vector field $\mathbf{E}(\widetilde{\mathbf{x}}|c)$. Specifically, for a data sample $\widetilde{\mathbf{x}}_q = (\mathbf{x}_q, 0)$ from $c$-th class, we sample noised sample $\widetilde{\mathbf{x}}$ via (10) and approximate $\mathbf{E}(\widetilde{\mathbf{x}}|c)$ by a neural network $f_\theta(\widetilde{\mathbf{x}}, c)$ with the following optimization function over parameters $\theta$:

$$\mathbb{E}_c \mathbb{E}_{\widetilde{\mathbf{x}}|c} || f_\theta(\widetilde{\mathbf{x}}, c) - \frac{\mathbf{E}(\tilde{\mathbf{x}}|c)}{||\mathbf{E}(\tilde{\mathbf{x}}|c)||_2} ||_2^2 \to \min_\theta.$$

We consider conditional generating task on the 32x32 CIFAR-10 dataset and demonstrate generated images over each class $c$ in Fig. 16

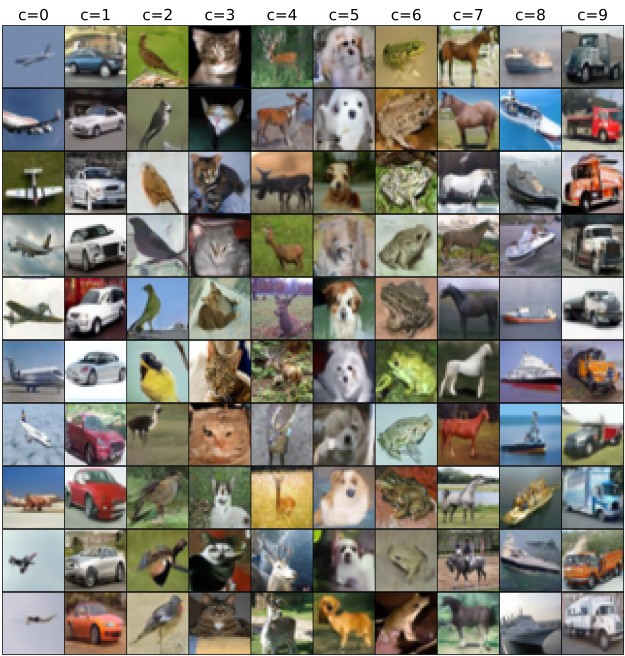

Figure 16: *Conditional Image Generation*: Samples obtained by conditional vector field $\mathbf{E}(\tilde{\mathbf{x}}|c)$ of IFM(ours) method on CIFAR-10 dataset for each class $c$.

## F    COMPARISON WITH FLOW MATCHING

Our IFM framework offers an important advantage compared to Flow Matching (Lipman et al., 2023; Liu et al., 2023; Tong et al., 2023): it enables **multi-sample estimation of the field**.

In particular, **our IFM method** approximates the normalized vector field with a neural network $f_\theta(\widetilde{\mathbf{x}})$, trained with the loss

$$\mathcal{L}_{\mathrm{IFM}} = \mathbb{E}_{\widetilde{\mathbf{x}}} \left\| f_\theta(\widetilde{\mathbf{x}}) - \frac{\mathbf{E}(\widetilde{\mathbf{x}})}{\|\mathbf{E}(\widetilde{\mathbf{x}})\|} \right\|_2^2,$$

which requires an estimate of the ground-truth vector field $\mathbf{E}(\widetilde{\mathbf{x}})$. The distribution over points $\widetilde{\mathbf{x}}$ at which the field is learned serves as a hyperparameter. Since the field $\mathbf{E}(\widetilde{\mathbf{x}})$ is represented using the superposition principle (7), we can estimate it by averaging over fields induced by $B$ batch samples (quark and anti-quark pairs) $\widetilde{\mathbf{x}}_q = (\mathbf{x}_q, 0)$ and $\widetilde{\mathbf{x}}_{\bar{q}} = (\mathbf{x}_{\bar{q}}, L)$:

$$\mathbf{E}(\widetilde{\mathbf{x}}) \approx \frac{1}{B} \sum_{i=1}^{B} \mathbf{E}_{\mathbf{x}_{q_i} \mathbf{x}_{\bar{q}_i}}(\widetilde{\mathbf{x}}),$$

where each $\mathbf{E}_{\mathbf{x}_{q_i} \mathbf{x}_{\bar{q}_i}}(\widetilde{\mathbf{x}})$ admits a closed form (see Appendix A.4). *Thus, we can use any available number of sample pairs $(\widetilde{\mathbf{x}}_q, \widetilde{\mathbf{x}}_{\bar{q}}) \sim \pi$—up to the entire dataset—to estimate the ground-truth field and reduce the variance of this Monte Carlo estimator.*

In contrast, the **Flow Matching (FM)** loss is

$$\mathcal{L}_{\mathrm{FM}} = \mathbb{E}_{t\in[0,1],\,(\mathbf{x}_0,\mathbf{x}_1)\sim\pi} \left\| \mathbf{v}_\theta(\mathbf{x}_t, t) - (\mathbf{x}_1 - \mathbf{x}_0) \right\|_2^2,$$

where $\mathbf{x}_t = t\mathbf{x}_1 + (1-t)\mathbf{x}_0$. The optimal vector field is $\mathbf{v}^*(\mathbf{x}_t, t) = \mathbb{E}[\mathbf{x}_1 - \mathbf{x}_0 \mid \mathbf{x}_t]$, but this conditional expectation is intractable to estimate via Monte Carlo because one cannot easily sample $\mathbf{x}_1, \mathbf{x}_0$ conditioned on $\mathbf{x}_t$. Thus, during training, one regresses $\mathbf{v}_\theta(\mathbf{x}_t, t)$ to its single-sample estimate

$$\mathbf{v}_\theta(\mathbf{x}_t, t) \approx \mathbf{x}_1 - \mathbf{x}_0.$$

*Therefore, FM estimates the ground-truth field at each point $\mathbf{x}_t$ using only one pair $(\mathbf{x}_0, \mathbf{x}_1) \sim \pi$, with no direct way to reduce the variance of this Monte Carlo estimate.*

We sum up differences between FM and our IFM in Table 6 .

|  | IFM (ours) | FM |
|---|---|---|
| Estimation of a field | Multi-sample: $\mathbf{E}(\widehat{\mathbf{x}})$ over B pairs $(\mathbf{x}_0, \mathbf{x}_1) \sim \pi$ | One-sample: $\mathbf{v}(\mathbf{x}_t)$ over one pair $(\mathbf{x}_0, \mathbf{x}_1) \sim \pi$ |
| Training volume | Any: $\widetilde{\mathbf{x}} = (\mathbf{x}, z) : z \in [0, L]$ | Restricted: $\mathbf{x}_t = t\mathbf{x}_1 + (1 - t)\mathbf{x}_0$ |

Table 6: The differences between our IFM and Flow Matching (FM).

