# OpenReview forum: "Interaction Field Matching: Overcoming Limitations of Electrostatic Models"
_ICLR.cc/2026/Conference — ICLR 2026 Poster_

### Official Review · Reviewer_PXv4 · 2025-10-16

**Soundness:** 3
**Presentation:** 3
**Contribution:** 3
**Rating:** 6
**Confidence:** 3

**Summary:**

This paper proposes a generative model based on EFM, which allows using general interaction fields beyond the existing electrostatic ones. The main theorem shows that IFM can recover the target distribution successfully. Experimental results show that it can outperform competitive models in image generation and image-to-image translation.

**Strengths:**

1. The mathematical results look sound. And it identifies and resolves some problems from previous EFMs.

2. The empirical results are solid, and they are able to demonstrate stronger performance than GANs and DDPMs.

**Weaknesses:**

1. Ablation studies are not presented in the main paper. It's hard to know what components contribute to the final model gain.

Running speed isn't provided. So we don't know whether this presented model is, in fact, scalable.

The resolution of the images used in this paper is quite low.

2. Presentation issues:

(1)  It is better to mark z=L line in Figure 2. It's not very clear now.
(2)  Equation 11 seems to be incomplete. Or maybe it means an SGD step. It's unclear.
(3) In Table 1, our->ours.

**Questions:**

1. Why does EFM get completely wrong results?

2. Why was this ODE better than a straight-forward linear ODE?

---

> ### Author Response · Authors · 2025-11-21
> **Answer to all questions**
>
> Dear reviewer, thank you for your kind words. We are pleased that you positively highlight our mathematical results and experimental performance. Below we comment on your questions.
>
> **(Q1) Running speed isn't provided. So we don't know whether this presented model is, in fact, scalable?**
>
> Please note that we have already provided the running speed of our method depending on batch size in Table 4 (see lines 1097-1101 of revised paper).
>
> **(Q2) The resolution of the images used in this paper is quite low?**
>
> We follow the common practice established in [1,2], we conduct our experiments in a 3x64x64 dimensional space to compare our approach with other papers.  However, in response to your request, we have also provide a more challenging image generation experiment on the 128x128 CelebA dataset. We describe this experiment in detail in the **newly added Appendix D** of the revised paper.
>
> **(Q3) Why does EFM get completely wrong results?**
>
> As our theoretical analysis suggests (see section 2.3), EFM has three crucial issues: backward-oriented field lines, line termination problem and training volume selection. This problematic behavior is already observable in toy experiments (see Figure 2) and obviously becomes more severe with increasing dimensionality when working with images (see Figures 9(b) and 10(b)).
>
>
>
> **(Q4) Ablation studies are not presented in the main paper. It's hard to know  what components contribute to the final model gain?**
>
> Compared to the most related prior work, EFM, we modify only a single component: the type of field. Our results demonstrate that IFM, which introduces a quark-based field, leads to an improvement over the original EFM. We have already addressed the reasons for EFM's limitations in our response to Question Q3.
>
> Regarding the ablation study, we have followed your request and included a detailed analysis of our method's hyperparameters in the **newly added Appendix C** of the revised paper.
>
> **(Q5) Why was this ODE better than a straight-forward linear ODE?**
>
> If we understand correctly, by "straightforward linear ODE" you are referring to the connection between our method and Flow Matching (FM). We would like to clarify, however, that FM is significantly different from our approach in several key aspects. We elaborate on these distinctions in the **newly added Appendix F** of the revised paper.
>
>
> **Concluding remarks**.
> We would be grateful if you could let us know if the explanations we gave have been satisfactory in addressing your concerns about our work. If so, we kindly ask that you consider increasing your score. We are also open to discussing any other questions you may have.
>
>
> **References**
>
> [1] Xu et. al., NeurIPS (2022). Poisson Flow Generative Models.
>
> [2] Lipman et. al., ICLR (2023). Flow Matching for Generative Modeling.

---

> > ### Comment · Reviewer_PXv4 · 2025-11-21
> > **OK but not great rebuttal**
> >
> > Hi authors,
> >
> > Thanks for your rebuttal. The reviewer still feels positive about this paper, but he decides to maintain the overall score and **decrease** the presentation score by one.
> >
> > The CelebA dataset 128 results are very bad, and no FID is reported. Very old methods can achieve much better results on this dataset (https://github.com/NVlabs/stylegan3). Also, it's not consistent with 64 results. However, the reviewer can tolerate more novel methods, though.
> >
> > Let's take the Q1 issue as an example, where the reviewer feels it's not as great as possible:
> >
> > (1) The speed experiment should be presented in the main paper, as it matters for the overall understanding of a generative model.
> >
> > (2) The speed results are not even referred to in the paper, nor in the revised paper.
> >
> > (3) Appendix B is referred to once in the revised paper: "We give technical details of the implementation of the experiments in Appendix B." Speed results are not implementation details. So this sentence remains inaccurate.
> >
> > (4) Speed results are not compared to the closest baselines.
> >
> > (5) Reporting speed results w.r.t. batch sizes is very confusing. Usually, it should be per image speed. Please refer to other standard generative AI papers and report correspondingly.
> >
> > (6) It's only reported in one resolution, while the scaling effects of the speed remain uncertain.
> >
> > Given all those points, this paper's writing does not match the highest standard of a generative AI paper.

---

> > > ### Author Response · Authors · 2025-11-26
> > > **Answer**
> > >
> > > Dear reviewer, we are grateful for your response and would like to comment on your questions further.
> > >
> > > **(Q1) The CelebA dataset 128 results [...] old methods can achieve much better results [...]**
> > >
> > > As per your request, we provided a very quick experiment on 128-resolution Celeba in our rebuttal. While we agree that our results here may be not SOTA, we simply achieved this *without any tuning* just by slightly modifying Celeba 64x64 code. We believe that the qualitative performance is reasonable in such a case. In fact, since our approach is based on novel principles, we believe that making our approach to reach peak performance, especially in high-dimensional images, may require additional tuning which is left for future research. We further emphasize that:
> > >
> > > - Achieving scalability and state-of-the-art (SOTA) performance on high resolutions typically requires an extensive and complex process of tailoring architectures and hyperparameters to a specific training pipeline. For instance, models like StyleGAN were originally tuned across a massive number of GPUs for an immense amount of time (**50 V100 GPU years for StyleGAN2** [1])  to obtain generalizable hyperparameters. Such resources far beyond our current means. Furthermore, our paper is conceptual in nature, not focused on achieving SOTA results.
> > >
> > > - Regarding widespread diffusion models. Note that the first diffusion models appeared in 2015 [2] and initially performed poorly, even on datasets like CIFAR. It took **4+ years of research**, involving the development of correct parameterizations and other techniques, to successfully scale them up. Therefore, we leave the tuning and refinement of our novel approach for future work. In this paper, we believe that have presented a sufficient set of experiments that effectively illustrate the underlying theory and methodology.
> > >
> > > In summary, given that our primary goal is to demonstrate a proof-of-concept for our novel methodology for learning generative models, we humbly think that our current experiments (e.g., CIFAR-10 32x32 and CelebA 64x64) benchmarks are already sufficient to fulfill this goal. In light of the above, we hope the reviewer agrees that our work makes a valuable conceptual contribution to the field.
> > >
> > > **(Q2) Questions related to speed of the method.**
> > >
> > > If we understand your series of questions correctly, you are requesting a careful comparison of the runtime during inference between our method and the baselines (**Comment 4**). As previously mentioned in Appendix B (lines 1093-1096), we have ensured an fair comparison between methods by using identical architectures, the same ODE-based inference procedure, and identically same number of function evaluations (NFE). Consequently, the runtime of our method is exactly **same** as that of the baselines, so it is a little bit unclear for us what exactly do you request.
> > >
> > > In response to **Comment 5** on per-image speed, we would like to pay your attention to Tables 2 and 4 of the revised paper, where the inference time for a batch size of 1 is already provided. Regarding **Comment 6**, we follow your request by reporting the runtime of our model on the generation of CelebA 128x128, CIFAR-10 and translation of MNIST and Winter2Summer. The results include both per-image inference time (batch size=1) and timings for various batch sizes with the blue color in Tables 2 and 4 of the revised paper.
> > >
> > > In response to  **Comments 1-3**, we have followed the reviewers' suggestions and added corresponding runtime descriptions to  **Sections 4.2** and  **4.3**. These sections discuss the inference speed of our method for both scenarios: noise-to-data and data-to-data.
> > >
> > >
> > > **Concluding remarks**. Following the modifications related to Q2, we uploaded the revised paper. We would be grateful if you could let us know if these updated we gave have been satisfactory in addressing your questions about our work.
> > >
> > >
> > > **References**
> > >
> > > [1] Karras  et. al., CVPR (2020). Analyzing and Improving the Image Quality of StyleGAN.
> > >
> > > [2] Sohl-Dickstein J. et. al., ICML (2015). Deep unsupervised learning using nonequilibrium thermodynamics.

---

### Official Review · Reviewer_gzz2 · 2025-10-29

**Soundness:** 4
**Presentation:** 4
**Contribution:** 3
**Rating:** 8
**Confidence:** 3

**Summary:**

The authors propose Interaction Field Matching (IFM), a generalization of electrostatic field matching. The authors further design a particular instance of IFM that solves several problems with EFM and show promising performance on several toy and image data transfer problems. The authors further provide extensive theory to show that their model provably transfers between distribution and provide the most general set of requirements for the interaction field to perform data transfer.

**Strengths:**

- The authors provide a strong theoretical contribution by generalizing EFM and providing the most general set of requirements for the interaction field to perform data transfer.
- The paper is very clearly written, intuitive, and easy to understand
- The proposed IFM solves several problems with EFM and achieves better performance on several tasks.
- The proposed IFM is even competitive with/outperforms diffusion/flow-matching methods.
- Overall, the paper has both very strong theoretical results and promising empirical studies

**Weaknesses:**

- This is somewhat of a minor point, but the authors only provide experiments on relatively "toy" problems. It would be more convincing if the authors included additional experiments on other, more challenging image generation tasks.
- The paper could use some additional discussion on diffusion/flow matching to situate the work in the broader context of generative modeling

**Questions:**

As someone less familiar with the IFM, I am curious what is the connection between IFM/EFM and flow-matching? Is it possible to generalize them with a single theory?

---

> ### Author Response · Authors · 2025-11-21
> **Answer to all questions**
>
> Dear reviewer! Thank you for your positive feedback and profound questions. Below we answer your comments.
>
> **(Q1)  The paper could use some additional discussion on diffusion/flow matching to situate the work in the broader context of generative modeling. As someone less familiar with the IFM, I am curious what is the connection between IFM/EFM and flow-matching? Is it possible to generalize them with a single theory?**
>
> We follow your request and add a detailed description of the distinctions between our approach and Flow Matching (FM). This comparison can be **found in the new Appendix F** of our revised paper. In short, there are notable differences that make it non-trivial to generalize them within a single theory. Please, take it for consideration.
>
> **(Q2)  This is somewhat of a minor point, but the authors only provide experiments on relatively "toy" problems. It would be more convincing if the authors included additional experiments on other, more challenging image generation tasks.**
>
> We follow the common practice established in [1,2] by conducting our main experiments in a 3x64x64 dimensional space to ensure a fair comparison with other approaches. However, in response to your request, we have also provide a more challenging image generation experiment on the 128x128 CelebA dataset. We describe this experiment in detail in the **newly added Appendix D** of the revised paper.
>
>
> **Concluding remarks.** We would be grateful if you could let us know if we have satisfactory answered your questions and addressed your concerns. We are also open to discussing any other questions you may have.
>
>
> **References**
>
> [1] Xu et. al., NeurIPS (2022). Poisson Flow Generative Models.
>
> [2] Lipman et. al., ICLR (2023). Flow Matching for Generative Modeling.

---

> > ### Comment · Reviewer_gzz2 · 2025-11-27
> > **Response to Author Comments**
> >
> > I thank the authors for their response and additional explanations in the updated appendix. I will maintain my score and recommend acceptance of this work as I believe it is a novel and interesting contribution to the field.

---

### Official Review · Reviewer_TnaZ · 2025-10-31

**Soundness:** 3
**Presentation:** 3
**Contribution:** 2
**Rating:** 4
**Confidence:** 2

**Summary:**

The paper proposes Interaction Field Matching (IFM), a physics-inspired method for data distribution transfer problems. In comparison to Electrostatic Field Matching (EFM), IFM generalized to arbitrary interactions between charges and doesn’t suffer the same problems with tracing backward-oriented field lines. The paper presents the main theorem that movement along interaction field lines provably transfers one data distribution to another. Experiments demonstrate that IFM performs comparably in image generation tasks to well-known approaches such as flow-based models and DDPM.

**Strengths:**

The paper is well written and easy to follow, with careful explanations of the limitations of EFM (e.g. the line termination problem in Figure 2). It creatively draws from ideas in physics to propose reasonable properties of interaction fields (e.g. the start/termination of lines and flux conservation). It is especially useful that IFM provably transfers one data distribution to the other. From the visualizations in Figure 9 and the numerical results in Table 1, it seems that IFM is comparable to other state-of-the-art methods for image generation (slightly outperformed by PGFM), illustrating that this could be a promising method for data to data transfer.

**Weaknesses:**

The main weakness of this work is that it seems to be a repacking of the Maximum Mean Discrepancy with a field-induced kernel. Thus, I am unsure of the novelty. Given two probability distributions $p(x), q(x)$ the MMD is the squared distance between their mean embeddings in a reproducing kernel Hilbert space with some kernel $k(x,x’)$. See [1]. From my understanding, in IFM, you are replacing the kernel $k(x,x’)$ with the interaction field (so this is essentially MMD with a field-induced kernel). The IFM loss is then the MMD$^2$. Thus, to me this represents a “repackaged” MMD in the language of fields, and not really a new generative principle. I believe one can derive this mapping to the empirical MMD$^2$ from Eq. 7/the IFM loss. It is essentially comparing distributions via pairwise kernel sums (so is a kernel-based MMD method, e.g. in section 3.2.3). Please correct my understanding if this is wrong, but I think it would be worth addressing the connection to MMD/other distribution alignment techniques.

The method relies on choosing an interaction field, but there aren’t many details provided about how one should choose this field/in which scenarios it may be beneficial to choose a certain field over others.

The IFM method performs comparatively well to other methods on image generation tasks but does not outperform significantly. It would be helpful to discuss specific scenarios where IFM could provide advantages over existing methods.

I’m not sure what the point of the image-to-image translation task is besides showing that IFM can do what EFM cannot. It seems relatively simple compared to standard benchmarks in image-to-image translation (e.g. Cityscapes).

**Questions:**

How is this related to kernel-based MMD methods (see above)? It might strengthen the paper to explicitly discuss the connection to MMD and clarify what additional value this perspective of field interactions brings.

Are there guidelines besides the general requirements for choosing the interaction field? Did the authors do ablations on different choices of interaction field?

Can the authors motivate why one would want to use IFM over existing generative frameworks?

[1] Gretton, A., Borgwardt, K. M., Rasch, M. J., Schölkopf, B., & Smola, A. J. A Kernel Two-Sample Test. Journal of Machine Learning Research (JMLR), 2012.

---

> ### Author Response · Authors · 2025-11-21
> **Answer to questions 1-3**
>
> Dear reviewer, thanks for your efforts and your thoughtful review. Please find answers to your questions and comments below.
>
> **(W1.1) The main weakness ... repacking of the Maximum Mean Discrepancy with a field-induced kernel. Thus, I am unsure of the novelty... [...]... Please correct my understanding if this is wrong**
>
> We believe that there might be a misunderstanding about the relation of our approach to MMD. Our method is not inherently related to the distribution features nor it compares the distributions through a loss. Here are the answers to the key points:
>
> **[...] The IFM loss is then the MMD$^{2}$. [...] It is essentially comparing distributions via pairwise kernel sums**
>
> MMD works by comparing the distributions. The closer the distributions are to each other, the smaller the MMD metric. For identical distributions the metric is zero. In the IFM approach, the loss function we minimize does not aim to reduce the distance between distributions. The whole idea of our framework is to use the interaction field produced by distributions and its lines to transport the probability mass from quark distribution to the antiquarks. The minimization just helps a neural network to learn this field. The minimization will proceed and the non-zero field found even for the identical distributions.
>
> **[...] From my understanding, in IFM, you are replacing the kernel $k(x,x')$ with the interaction field (so this is essentially MMD with a field-induced kernel) [...]**
>
> In our case, for a pair $q,\overline{q}$ of quark-antiquark located at points $(x_0,x_1)$, the field at an arbitrary point $x'$ is given by $E_{q,\overline{q}}(x')$. It is a function of **three variables** $x_0,x_1,x'$ and can be written as $E_{q,\overline{q}}(x')=E(x'|x_0,x_1)$. In turn, in MMD, when one considers kernels $k$, they are functions of **two variables**, e.g., $k(x_0,x')$ or $k(x_1,x')$. Thus, it is unclear how our interaction field can be used as a substitute for the kernel in MMD.
>
> *The said above shows that there is no clear connection of our approach to MMD, and we kindly ask to reconsider evaluation of our paper.*
>
> **(Q2) The method relies on choosing an interaction field [...] Are there guidelines besides the general requirements for choosing the
> interaction field? Did the authors do ablations on different choices of
> interaction field?**
>
>
> To choose a specific field, one needs to design a construction that satisfies several requirements - the interaction field line must start at the quark and end at the antiquark, satisfy flux conservation and the generalized superposition principle (see the section 3.2). To satisfy the first and the last requirement, it is sufficient to define the field for an elementary quark/antiquark pair construction and then apply it to the entire distribution according to formula (7).
>
> To satisfy the flux conservation requirement, it is important to note that any flow tube must narrow to a point as it approaches a charge. Consequently, the cross-sectional area (which is proportional to $\sim \sigma(z)^D$) must also decrease. Flow conservation can only be maintained by a proportional increase in the field strength (see Fig. 13 of the revised paper).
>
> This represents the general recipe for constructing and selecting an IFM field. While a comprehensive comparison of different field realizations across various scenarios remains an interesting direction for future work, our proposed IFM field offers distinct advantages over the electrostatic approach. Specifically, it successfully addresses key limitations of EFM, including backward-oriented field lines, the line termination problem, and training volume selection challenges discussed in the section 2.3.
>
> We also note, that in the revised version of our paper, we include a comprehensive ablation study in Appendix C that examines our model's sensitivity to the hyperparameters $L$, $\sigma_0$, and $d$.
>
> **(Q3) Can the authors motivate why one would want to use IFM over
>  existing generative frameworks?**
>
> Our IFM framework offers  primary advantages over existing generative models.
>
>
>
> Compared to **PFGM** [1]: Our method is more flexible as it supports both data-to-data and noise-to-data trajectories, whereas PFGM is limited to noise-to-data.
>
> Compared to **Diffusion Models** [2]: IFM's straighter field lines property, as predicted by our theory, potentially allows faster generation (lower NFE) due to straighter trajectories.
>
> Advantages over **Normalizing Flows** [3]: Our framework is more expressive by design, as it is free from the requirements of invertible functions and computationally efficient Jacobian calculations that limit normalizing flows.
>
> Regarding the comparison with **Flow Matching** [4]. As this method may look deceptively close to our method, we discuss the differences and advantages in the **newly added Appendix F** of our revised paper. Please, take it for consideration.

---

> ### Author Response · Authors · 2025-11-21
> **Answer to question 4**
>
> **(Q4)  I’m not sure what the point of the image-to-image translation task is besides showing that IFM can do what EFM cannot. It seems
> relatively simple compared to standard benchmarks [...].**
>
> We generally agree with the reviewer, but the main point of these experiments is to demonstrate the proof of concept. We believe that the provided experiments fullfill this purpose.
>
>
> **Concluding remarks.** We thank the reviewer for the raised concerns and questions. We hope that we have properly addressed them. If so, we kindly ask the reviewer to consider rising the score and confidence. We are ready to continue the discussion of the existing questions, and we are also happy to answer new ones.
>
>
> **References**
>
> [1] Xu et. al., NeurIPS (2022). Poisson Flow Generative Models.
>
> [2] Ho et. al., NeurIPS, (2020). Denoising Diffusion Probabilities Models.
>
> [3] Rezende et. al., ICML, (2015). Variational Inference with Normalizing Flows.
>
> [4] Lipman et. al., ICLR (2022). Flow Matching for Generative Modeling.

---

> > ### Comment · Reviewer_TnaZ · 2025-11-26
> > **Thank you for the responses/clarification**
> >
> > Thanks for the responses! Apologies for the misunderstanding about the MMD (this is not my main field of expertise), and thanks for the clarification. I have updated my score accordingly to 6. In my view, the paper clearly presents a method that overcomes the challenges of EFM. It is still somewhat unclear to me for which tasks one would want to use IFM as opposed to other frameworks. E.g. I wonder if there are specific datasets or modeling tasks that would be well-tailored to IFM?

---

> > > ### Author Response · Authors · 2025-12-01
> > > **Answer**
> > >
> > > We thank the reviewer for the positive assessment and for the questions regarding the specific applicability of IFM.
> > >
> > > To answer your question, we highlight that unpaired learning problems are often addressed through the lens of Optimal Transport (OT), which seeks direct, minimal-cost mappings between distributions using a transport plan [1,2,3]. The theoretical foundation of our approach (IFM) similarly leverages a transport plan to estimate a vector field, resulting in straight trajectories along which data is transformed (see lines 221-252). This conceptual similarity suggests that the future development and application of IFM are especially promising in this context of unpaired learning and OT. Although seemingly related methods such as OT-CFM [4] exist, they have inherent differences from our method, which may make IFM preferable in certain settings; see our discussion in Appendix F. Further investigation of this aspect is a promising avenue for future research.
> > >
> > > **References**
> > >
> > > [1]  Zhu J. Y. et al. "Unpaired image-to-image translation using cycle-consistent adversarial networks"
> > >
> > > [2]  Korotin A. et al. "Neural optimal transport"
> > >
> > > [3]  Liu X. et al. "Flow straight and fast: Learning to generate and transfer data with rectified flow"
> > >
> > > [4]  Tong A. et al. "Conditional flow matching: Simulation-free dynamic optimal transport"

---

### Official Review · Reviewer_KeQQ · 2025-11-02

**Soundness:** 3
**Presentation:** 2
**Contribution:** 3
**Rating:** 6
**Confidence:** 2

**Summary:**

The paper proposes Interaction Field Matching (IFM), a novel physics-inspired framework for data generation and transfer designed to overcome limitations of existing Electrostatic Field Matching (EFM) models. The authors identify issues with EFM such as curved field lines, extension beyond target regions, and insufficient coverage of the target distribution. IFM addresses these problems by introducing a new field structure that results in nearly straight field lines between planes, prevents extension beyond z>L, and adequately covers the entire target distribution. The paper provides theoretical foundations for the method, including proofs of flow conservation properties, and validates it through experiments on multiple datasets. Results show that IFM performs well on datasets like 64×64 CelebA where EFM fails. Additionally, IFM demonstrates competitive performance in generation quality compared to state-of-the-art methods such as PFGM, Flow Matching, DDPM, and StyleGAN.

**Strengths:**

1. The paper proposes a novel physics-inspired framework that addresses key limitations of existing EFM methods. IFM produces nearly straight field lines and effectively covers the target distribution, which is significant both theoretically and practically.
2. The authors provide solid theoretical foundations, including proofs of flow conservation properties and mathematical analysis of field line behavior. Theorems and lemmas in the appendix guarantee the correctness of the method.
3. The experimental design is comprehensive, comparing not only with EFM, PFGM, and PFGM++ but also with modern flow-based methods (Flow Matching), diffusion-based methods (DDPM), and adversarial approaches (StyleGAN), demonstrating IFM's effectiveness.
4. Figure 3 clearly illustrates IFM's advantages over EFM, visually explaining why IFM better handles data transfer problems. This visualization is extremely helpful for understanding the method's improvements.
5. The authors honestly discuss the limitations of their method, particularly the numerical precision issues that may arise in high dimensions (1/σ(z)^D potentially producing values close to machine precision). This transparency enhances the paper's credibility.

**Weaknesses:**

1. Although the paper mentions potential numerical precision issues in high dimensions, it lacks systematic analysis of their practical impact. In real applications, when dimension D is large, this issue could cause algorithm instability or failure, but the paper does not provide solutions or mitigation strategies.
2. The experimental section only presents qualitative results (such as Figures 9a and 9b) without quantitative evaluation metrics. In the field of generative models, standard metrics like FID and IS are crucial for objective performance comparison, but the paper does not report these.
3. The paper does not provide detailed analysis of IFM's computational complexity and runtime. Compared to existing methods, does IFM introduce significant computational overhead? This is crucial for practical applications, but the paper does not provide relevant information.
4. Insufficient parameter sensitivity analysis. IFM likely depends on key parameters (such as distance L, function σ(z), etc.), but the paper does not systematically study how these parameters affect performance. Figure 4 shows results for different L values, but lacks analysis of other parameters.
5. The comparison with existing methods is not comprehensive. While the paper mentions comparisons with PFGM and PFGM++, it does not detail the specific theoretical and implementation differences between IFM and these methods, or why IFM performs better in certain cases.
6. The paper does not explore IFM's applicability to other data types (such as text, audio, or graph-structured data). While the focus is on image generation, discussion of extension to other data modalities would enhance the method's generality.

**Questions:**

1. How does the small value problem from 1/σ(z)^D practically affect algorithm performance in high dimensions (e.g., D>100)? Have you tried any numerical stability techniques (such as log-space calculations) to mitigate this issue?
2. Figure 4 shows the impact of different L values on results but does not explain how to select the optimal L value. In practical applications, is there an automatic method to determine L? Is there any relationship between L and dataset characteristics?
3. How does IFM's training and sampling efficiency compare with other generative models? Could you provide quantitative comparisons with methods like PFGM and DDPM in terms of computational time and memory usage?
4. How is the "crucial flow conservation property" mentioned in Appendix A.4 ensured? Could you explain in more detail why IFM's field structure maintains this property while EFM cannot?
5. Can the IFM framework be extended to conditional generation tasks? For example, could class information or text descriptions be incorporated into the field structure for conditional image generation?

---

> ### Author Response · Authors · 2025-11-21
> **Answer to question 1-3**
>
> Dear Reviewer, thank you for your positive and encouraging feedback. We are delighted that you found our theoretical foundations and experimental design to be comprehensive. Below we respond to your specific questions
>
> **(Q1) [...] numerical precision issues in high dimensions [...] their practical impact [...] when dimension D is large [...] How does the small value problem from $1/\sigma(z)^D$ practically affect algorithm performance in high dimensions (e.g., D>100)? Have you tried any numerical stability techniques [...]?**
>
> Thank you for your question. As we realized during the rebuttal, our IFM method **does not have** this limitation. We learn a **normalized** field  $\textbf{E}(\cdot)/||\textbf{E}(\cdot)||$, and it overcomes the dependence on $1/\sigma(z)^{D}$, see below.
>
> Let $\widetilde{\textbf{x}} \in \mathbb{R}^{D+1}$ be a point where we estimate the normalized vector field using $B$ sampled pairs of quarks and anti-quarks. In accordance with the superposition principle (see eq.(7)), the resulting field is obtained as the average of $B$ independent fields $E(x_{\perp}^{(i)},z)\textbf{n}(x_{\perp}^{(i)},z)$ (see eq.(54) in Appendix A.4) from each pair, where $z$ is the component of $\widetilde{\textbf{x}}$ along the $z$-axis and $x_{\perp}^{(i)}$ is the distance between $\widetilde{\textbf{x}}$ and the axis connected  $i$-th pair of quarks and anti-quarks:
> $$\frac{\textbf{E}(\widetilde{\textbf{x}})}{||\textbf{E}(\widetilde{\textbf{x}})||} =  \sum_{i=1}^{B}\frac{\exp(-\frac{x_{\perp}^{(i)}}{2\sigma(z)^{2}})}{\cancel{\sigma(z)^{D}}}\textbf{n}(x_{\perp}^{(i)},z) \Bigg/ \frac{1}{{\cancel{\sigma(z)^{D}}}}  ||\sum_{i=1}^{B}\exp(-\frac{x_{\perp}^{(i)}}{2\sigma(z)^{2}}) \textbf{n}(x_{\perp}^{(i)},z)||.$$
>
>
> Given that $\sigma(z)^{D}$ multiplier is common to all $B$ terms, it can be factored out of the summation. The same applies to the denominator. Hence, the term cancels out completely.
>
> Thanks to bringing this to our attention, we removed this issue from the limitations in revised paper.
>
> **(Q2) The experimental section only presents qualitative results (such as Figures 9a and 9b) without quantitative evaluation metrics. In the field of generative models, standard metrics like FID and IS are crucial for objective performance comparison, but the paper does not report these.**
>
> We think that there might be a misunderstanding. *We have already provided quantitative evaluation metrics in the main text of our original submission*. For image generation, we employ the FID score (Table 1, lines 432-435 of revised paper), and for image-to-image translation, we use the CMMD metric (Table 3, lines 486-490 of the revised paper). Based on these presented metrics, our IFM method outperforms other methods on the Winter$\to$Summer and the colored '2'$\to$colored '3' setups as measured by CMMD, while demonstrating competitive performance on the image generation task as measured by FID.
>
> **(Q3)  The paper does not provide detailed analysis of IFM's computational complexity and runtime. Compared to existing methods, does IFM introduce significant computational overhead? [...] the paper does not provide relevant information. [...] Could you provide quantitative comparisons with methods like PFGM and DDPM in terms of computational time and memory usage?**
>
>
> Please note that we have already provided in the original submission the runtime of our method depending on batch size in Table 2 and Table 4 (see lines 450-454 and 498-500 of the revised paper) and the peak GPU memory usage on an NVIDIA A100 GPU (see lines 447-449 of the revised paper). Compared to EFM, our approach uses a different type of field and introduces the transport plan. Just like EFM, it maintains the U-Net architecture, ODE Euler sampling scheme, and other technical components shared by other models (PFGM, PFGM++, DDPM, FM); therefore, it incurs no computational overhead.

---

> ### Author Response · Authors · 2025-11-21
> **Answer to questions 4-6**
>
> **(Q4)  [...] parameter sensitivity analysis [...] distance L, function $\sigma(z)$, etc.) [...] Figure 4 shows results for different L values, but lacks analysis of other parameters. Figure 4 shows the impact of different L values on results but does not explain how to select the optimal L value [...] is there an automatic method to determine L? Is there any relationship between L and dataset characteristics?**
>
> Our IFM realization is defined by the following hyperparameters: the distance $L$ between plates, the string width $\sigma_0$, and the distance $d$ over which field lines curve toward the charges. We choose these parameters based on the following ideas:
>
> 1) The parameter $\sigma_0$, as noted in the previous question, cancels out when computing the *normalized* field during training. Nevertheless, for numerical considerations, we usually choose $\sigma_0 = 1$.
>
> 2) The distance $d$ should not be chosen too short—this complicates data translation via the ODE due to the high curvature of the field lines in the region $z \in [0, d]\cup [L-d, L]$ . In practice, we usually use $d \in [0.1L,\ 0.5L]$.
>
> 3) Finally, as you correctly noted, the distance $L$ does not significantly impact translation quality in our method. This is different from EFM, where making $L$ too large significantly worsens the results. Our IFM realization is specifically designed to reduce this dependency through straight field segments in the region $z \in [d, L-d]$, where ODE integration follows straight lines. In practice, we set $L$ to be on the order of the data standard deviation:
> $L \sim \sqrt{D_{\mathbb{P}}}$ or $\sqrt{D_{\mathbb{Q}}}$, i.e., proportional to the standard deviation of the source or target data distribution.
>
> Following your question, we conducted an ablation study investigating the dependence on the hyperparameter $d$. The results show only weak dependence on this parameter. In the revised paper, you can find CelebA 64x64 generation results with different values of $d$ (see Appendix C, Fig. 14).
>
> **(Q5) The comparison with existing methods is not comprehensive. While the paper mentions comparisons with PFGM and PFGM++, it does not detail the specific theoretical and implementation differences between IFM and these methods, or why IFM performs better in certain cases.**
>
> As we mentioned in Appendix B and answer to your Q3 above, there are **no principal implementation differences** between our method and PFGM/PFGM++, because our method IFM shares the same U-net neural architecture, ODE sampling scheme, and implementation tools for competitive all methods (EFM, PFGM, PFGM++, FM, DDPM). As a result, our approach has comparable results with PFGM/PFGM++ on image generation setup (see section 4.2).
>
> However, the **methodological**  advantage of our method is its suitability for both noise-to-data and data-to-data scenarios, whereas PFGM and PFGM++ are confined to the noise-to-data case.
>
>
> **(Q6) The paper does not explore IFM's applicability to other data types (such as text, audio, or graph-structured data).**
>
> We follow the standard benchmarking [1, 2, 3] of generative modeling by evaluating our method on image generation and translation tasks. In principle, our method can be applied to audio generation tasks in exactly the same way as to image tasks because audio is typically represented as spectrograms, which are images in fact. As for text and graph-structured data, being discrete data types, our method is not directly applicable to them, as they require specialized discrete generative models whose development is beyond the scope of this paper. Nevertheless, this question is an interesting and promising direction for future work.

---

> ### Author Response · Authors · 2025-11-21
> **Answer to questions 7-8**
>
> **(Q7) How is the "crucial flow conservation property" mentioned in Appendix A.4 ensured? Could you explain in more detail why IFM's field structure maintains this property while EFM cannot?**
>
> Perhaps there is a misunderstanding. In fact, both EFM and our proposed IFM satisfy the flux conservation property (see our Example 3.2, where we prove that EFM is a special case of our IFM).
>
> The need to generalize EFM arises because the electrostatic approach has certain drawbacks – such as backward-oriented field lines, the line termination problem, and training volume selection – as we discussed in detail in the section 2.3 of our paper. To overcome these drawbacks, we first theoretically generalized EFM, and then, motivated by the strong interaction of elementary particles and aiming to overcome EFM's limitations, we devised a specific realization of the new IFM field.
>
> A formal proof that our IFM realization also conserves flux can be found in Appendix A.5 (Theorem A.8). It turns out that a crucial element in the proof is the factor $1/\sigma(z)^D$ in the definition of the IFM field (see Eq. (54)), where $D$ is the dimensionality of the space and $\sigma(z)$ is the string width. The intuition behind this factor can be explained as follows: any flow tube must narrow to a point as it approaches a charge. Consequently, the cross-sectional (which is proportional to $\sim \sigma(z)^D$) area must also decrease. Flux conservation can only be maintained by a proportional increase in the field strength (see Fig. 13 of the revised paper). In simpler terms, the closer one is to a charge, the stronger the field must be.
>
> We have added this intuitive explanation in the revised paper after the proof to improve the clarity.
>
> **(Q8) Can the IFM framework be extended to conditional generation tasks? For example, could class information or text descriptions be incorporated into the field structure for conditional image generation?**
>
> Yes, of course. We followed your request and provide  the corresponding experiment of conditional Image generation on CIFAR-10 dataset. This experimental setup is described in the newly added Appendix E. Please, take it for consideration.
>
>
>
> **Concluding remarks**. We thank the reviewer one more time for the interesting and important questions. Please respond to our post to let us know if the clarifications above suitably address your concerns about our work. We are happy to address any remaining points during the discussion phase; if the responses above are sufficient, we kindly ask that you consider raising your score and confidence.
>
> **References**
>
> [1] Ho et. al., NeurIPS, (2020). Denoising Diffusion Probabilities Models.
>
> [2] Lipman et. al., ICLR (2022). Flow Matching for Generative Modeling.
>
> [3] Xu et. al., ICML (2023). PFGM++: Unlocking the Potential of Physics-Inspired Generative Models.

---

### Author Response · Authors · 2025-11-21
**General response**

Dear reviewers, we want to express our gratitude for dedicating your valuable time to review our paper. Your feedback means a lot to us, and we are delighted that many of you found the theoretical and methodological contributions of our work to be significant and experiments to be sufficient.


**Revised paper:** In the revised manuscript, we have included several new appendices to address the reviewers' comments:

- **Appendix C**: An ablation study of our method IFM ( *for Reviewers KeQQ, PXv4* ).
- **Appendix D**: More challenging Image generation on CelebA 128×128 ( *for Reviewers gzz2, PXv4* ).
- **Appendix E**: Conditional Image generation on CIFAR-10 ( *for Reviewer KeQQ* ).
- **Appendix F**: Discussion of differences between our method (IFM) and Flow Matching (FM) ( *for Reviewers gzz2, TnaZ* ).

**All the edits are highlighted with the blue color.** Please take a look.

---

### Meta-Review · Area_Chair_QR43 · 2026-01-09

**Summary:**

The reviewers generally agree that the paper presents a meaningful theoretical generalization of Electrostatic Field Matching and that the proposed Interaction Field Matching framework is mathematically sound and well motivated. The strongest positive signal comes from the clarity of the theory and the fact that IFM resolves several known failure modes of EFM, which multiple reviewers found convincing. The main concerns that shaped the decision were instead centered on novelty framing and empirical strength. In particular, some reviewers questioned whether IFM constitutes a genuinely new generative principle or is better understood as a reformulation of existing ideas, and others felt that the experiments—while adequate for proof of concept—were limited in scope and difficulty relative to the current standards in generative modeling. Presentation and evaluation issues, including ablations, runtime reporting, and the quality of higher-resolution results, also contributed to hesitation about placing the paper more strongly above the acceptance threshold.

**Reviewer Concerns:**

The rebuttal was effective in addressing several of the most important concerns. Most notably, the authors provided a clear and convincing clarification regarding the relationship between IFM and MMD, resolving what turned out to be a misunderstanding rather than a fundamental flaw, and this directly led to at least one reviewer revising their assessment upward. The authors also added substantive new material, including ablation studies, conditional generation experiments, a discussion of differences from Flow Matching, and additional higher-resolution experiments. These changes improved both clarity and completeness. That said, not all concerns were fully resolved. The added CelebA 128×128 results are qualitatively weak and lack standard quantitative evaluation, which limits their impact, and the runtime and scalability discussion, while expanded, still does not reach the level of clarity and rigor typically expected in top-tier generative modeling papers. In addition, although the authors articulated plausible advantages of IFM, the paper still stops short of clearly delineating concrete application regimes where IFM is the method of choice over strong existing alternatives.

**Reviewer Scores:**

Reviewer KeQQ provided a marginally positive score with concerns centered on numerical stability, evaluation metrics, scalability, and parameter sensitivity. These concerns were largely addressed in the rebuttal and revised manuscript, but they do not appear to stem from a fundamental misunderstanding of the method. In the absence of further discussion from the reviewer, the AC believes it is most likely that the score would remain unchanged, with the rebuttal serving to reinforce rather than materially alter the original assessment. Reviewer TnaZ initially raised a novelty concern based on a perceived equivalence to MMD; this was a clear factual misunderstanding that was directly addressed, and the reviewer participated in the discussion and explicitly updated the score upward to a clear accept. Reviewer gzz2 was consistently positive and assigned a strong acceptance score; the reviewer participated in the discussion and explicitly stated that their score would be maintained. Reviewer PXv4 raised concerns regarding experimental presentation, runtime reporting, and the quality of higher-resolution results. Although the authors made revisions and provided additional explanations, the reviewer participated in the discussion and explicitly maintained their overall score while lowering the presentation subscore, indicating that the rebuttal did not resolve their concerns to a degree that would change the overall evaluation.

---

### Decision · Program_Chairs · 2026-01-26

Accept (Poster)